# Multilevel neurium-mimetic individualized graft via additive manufacturing for efficient tissue repair

Lingchi Kong [1,5], Xin Gao[2,5], Xiangyun Yao[1,3,5], Haijiao Xie [4], Qinglin Kang [1], Wei Sun [2] ✉, Zhengwei You [2] ✉, Yun Qian [1,3] ✉ & Cunyi Fan [1,3] ✉

Complicated peripheral nerve injuries or defects, especially at branching sites, remain a prominent clinical challenge after the application of different treatment strategies. Current nerve grafts fail to match the expected shape and size for delicate and precise branched nerve repair on a case-by-case basis, and there is a lack of geometrical and microscale regenerative navigation. In this study, we develop a sugar painting-inspired individualized multilevel epi-/peri-/endoneurium-mimetic device (SpinMed) to customize natural cues, featuring a selectively protective outer sheath and an instructive core, to support rapid vascular reconstruction and consequent efficient neurite extension along the defect area. The biomimetic perineurium dictates host-guest crosslinking in which new vessels secrete multimerin 1 binding to the fibroin filler surface as an anchor, contributing to the biological endoneurium that promotes Schwann cell homing and remyelination. SpinMed implantation into rat sciatic nerve defects yields a satisfactory outcome in terms of structural reconstruction, with sensory and locomotive function restoration. We further customize SpinMed grafts based on anatomy and digital imaging, achieving rapid repair of the nerve trunk and branches superior to that achieved by autografts and decellularized grafts in a specific beagle nerve defect model, with reliable biosafety. Overall, this intelligent art-inspired biomimetic design offers a facile way to customize sophisticated high-performance nerve grafts and holds great potential for application in translational regenerative medicine.

Peripheral nerve injury is a common and destructive problem in clinical practice. It is primarily characterized by slow revascularization, irreversible axon damage and mismatching, and severe sarcopenia, all of which eventually lead to sensory or motor disability in patients[1,2].

Currently, the most ideal clinical outcomes are expected to be achieved by simple end-to-end sutures for tension-free nerve gaps, whereas autografts remain the gold standard for critical defects (longer than 4 cm in humans)[2–4]. Although autografts mostly meet the

[1]Department of Orthopedics, Shanghai Sixth People's Hospital Affiliated to Shanghai Jiao Tong University School of Medicine, 200233 Shanghai, China. [2]State Key Laboratory for Modification of Chemical Fibers and Polymer Materials, College of Materials Science and Engineering, Donghua University, Research Base of Textile Materials for Flexible Electronics and Biomedical Applications (China Textile Engineering Society), Shanghai Engineering Research Center of Nano-Biomaterials and Regenerative Medicine, 201620 Shanghai, China. [3]Shanghai Engineering Research Center for Orthopaedic Material Innovation and Tissue Regeneration, 201306 Shanghai, China. [4]Hangzhou Yanqu Information Technology Co.Ltd., 310003 Hangzhou, China. [5]These authors contributed equally: Lingchi Kong, Xin Gao, Xiangyun Yao. ✉e-mail: weisun@dhu.edu.cn; zyou@dhu.edu.cn; lollipopcloudland@foxmail.com; sakio@sjtu.edu.cn; cyfan@sjtu.edu.cn

requirements for peripheral nerve regeneration (PNR), including high biocompatibility and natural structural composition, limitations such as limited donor availability, secondary damage, and size mismatching are still prominent and restrict their clinical application and therapeutic outcomes, especially in sophisticated nerve defect scenarios[5,6]. Tissue engineering strategies in recent years have been fully developed and applied in the area of PNR, yet a majority of them fail to achieve delicate and individualized mimicry of the native structure and, therefore, lead to unsatisfactory outcomes[7,8]. An ideal nerve graft is expected to guide geometrical and microscopic neurite outgrowth for efficient neural repair to alleviate the therapeutic dilemma.

Previous studies have reported a number of biomimetic nerve guidance conduit (NGC) types fabricated by electrospinning, gas foaming, and solvent casting to drive neural regrowth following injury[9,10]. However, these strategies are often difficult to translate and apply in complex clinical cases due to ambiguous bioactive components and difficulty in individualized manufacturing (e.g., producing specific architectures and mechanics). Thus, architectural simulation may be a promising direction for biomimetic nerve graft development. The natural nerve fiber is supported by multilevel neurium, epineurium, perineurium, and endoneurium, which contribute to a nerve repair platform[11–13]. Therefore, the neurium-mimetic fabrication of grafts, similar to allogeneic acellular nerves, could be used to manufacture grafts with similar architecture. Irregular nerve defects need grafts with unique shapes and sizes including nonuniform morphologies or bifurcations, instead of common autografts[14–17]. Regarding the difficulty in the accurate design and fabrication of nerve grafts with multiple branches, it is preferable to utilize advanced techniques for one-step three-dimensional (3D) fabrication, such as a form of additive manufacturing combined with digital imaging, for ideal graft production[18,19]. To achieve "precision medicine", another issue is integrating biomimetic layouts into grafts based on neural structures and the regenerative microenvironment to induce injured nerve fiber extending, for which detailed architectural designs and accessible fabrication techniques are needed. Precise nerve fiber guidance by biomimetic graft is undoubtedly helpful for promoting efficient neural regrowth. Therefore, individualized biomimetic strategies might hold promise for clinical translation and wide application.

Achieving an optimized microscale design calls for a full understanding of the spatiotemporal patterns of various regenerative units and essential characteristics of the microenvironment. Rapid reconstruction of structural integrity and microenvironmental homeostasis is crucial to optimizing reparative outcomes, as delayed surgery will result in significant functional loss[20]. The microenvironment during PNR is a dynamic landscape formed by biophysical and biochemical cues that regulate tissue regrowth and cell behaviors through direct interactions and paracrine pathways[9,21]. The clearance of myelin debris, vascular involvement, and Schwann cell (SC) remyelination together contribute to restoring the microenvironmental homeostasis of peripheral nerves, during which new vessel formation may be key for nourishing the microenvironment in a paracrine manner[22–24]. For instance, newly formed vessels contribute to functional cell activities during tissue regeneration through nutrient delivery and microenvironmental regulation[25]. Although there have been numerous studies on vascularization in PNR focusing on vessel-related nutrient supply and waste discharge[26–28], the paracrine modulation of vessels remains an understudied field. Moreover, many studies have concentrated on regenerative vessels within the epineurium, neglecting the indispensable role of intraneural vessels[29,30]. It is accepted that topological cue-induced vascular reconstruction guides regrowth direction (orientation or other well-organized cues) and provides microenvironmental homeostasis and a niche for neural tissue formation. Regarding the "nerve functional unit" composed of the axon/myelin and intraneural vessels, studies should focus on investigating the regrowth pattern of the intraneural vessel network within regenerative peripheral nerves and the paracrine modulation of myelin sheath formation and neurite outgrowth. These facts suggested that an intelligent nerve graft could be used to link biomaterials (host) and regenerative "nerve functional units" (guest) for maximal tissue regeneration efficiency, and after repair, allow the host-guest relationship to evolve such that fresh tissue (host) is integrated into the organism without biomaterial degradation products (guest) to achieve functional restoration.

In the present study, we hypothesize that optimized multilevel neurium-mimetic architectures, achieved by additive manufacturing and phase separation, could provide appropriate physical chambers and guide the oriented growth of intraneural vessels and that the well-formed intraneural vasculature would further contribute to the endoneurium, maintaining the microenvironmental homeostasis of neural tissue through paracrine function. The "vessel-myelin-axon" could be regarded as a "functional unit" of the microstructure expected to promote efficient neural regeneration. Accordingly, a sugar painting-inspired individualized multilevel epi-/peri-/endoneurium-mimetic device (SpinMed) was designed and applied in various nerve defect models.

## Results

### Performance of SpinMed

To ensure the adaptivity of the graft to the recipient site, sugar painting, a traditional art form in China, has inspired the biomimetic design of individualized grafts combined with phase separation, the raw material of which, sugar or caramel, confers plasticity and biosafety to the innovative nerve grafts (Fig. 1a). Regarding the indispensable roles of the neurium in microenvironment maintenance for neural growth, we constructed a multilevel neurium-mimetic architecture to mimic native epineurium/perineurium, offering structural support and a biomechanical microenvironment, and to induce the formation of an endoneurium-mimetic functional unit.

In clinical practice, we found that areas of the posttraumatic PNR were often destroyed following cicatricial fibrosis, as identified in 23 patients, due to the weak epineurium, making it difficult to keep out adjacent tissue (Supplementary Table 1). Single-cell RNA sequencing of human specimens revealed that smooth muscles actin α (αSMA)-positive muscle cell or pericyte invasion into the regenerative space primarily contributed to retarded neural regrowth (Supplementary Fig. 1; Supplementary Fig. 2a), and this result was similar to that of a previous study[31]. Linear regression analysis showed that intraneural fibrosis, especially in αSMA⁺ areas, significantly compromised regenerative nerve tissue (Supplementary Fig. 2b, c). Further experiments using an αSMA-tk mice nerve crush model demonstrated that deletion of αSMA⁺ cells restored, at least in part, neurite regrowth (Supplementary Fig. 2d–g). Consequently, epineurial design is expected to protect regenerative nerves from physical invasion and biological suppression of adjacent muscles and cells. The biomimetic epineurium was fabricated via phase separation following precise additive manufacturing (Supplementary Fig. 3a), where the epineurium with a hydrophobic surface and pores of ~1 μm in diameter, provided pericyte protection. The biomimetic perineurium filling the epineurial sheath was very similar to the natural perineurium, potentially providing physical interfaces for neural repair and supporting intraneural angiogenesis in an axial direction. Notably, the endoneurium, at the micrometer scale, is difficult to precisely mimic using current tissue engineering methods; thus, we employed a biological induction strategy involving host−guest crosslinking to facilitate spontaneous endoneurial formation.

Based on the above designs, SpinMed grafts with various sizes and shapes were fabricated and submitted to examinations (Fig. 1b−d). Morphological detection showed differential outer and inner interfaces (Fig. 1e), along with a notable distinction in hydrophilicity, as observed through water contact angle measurements; this feature, at

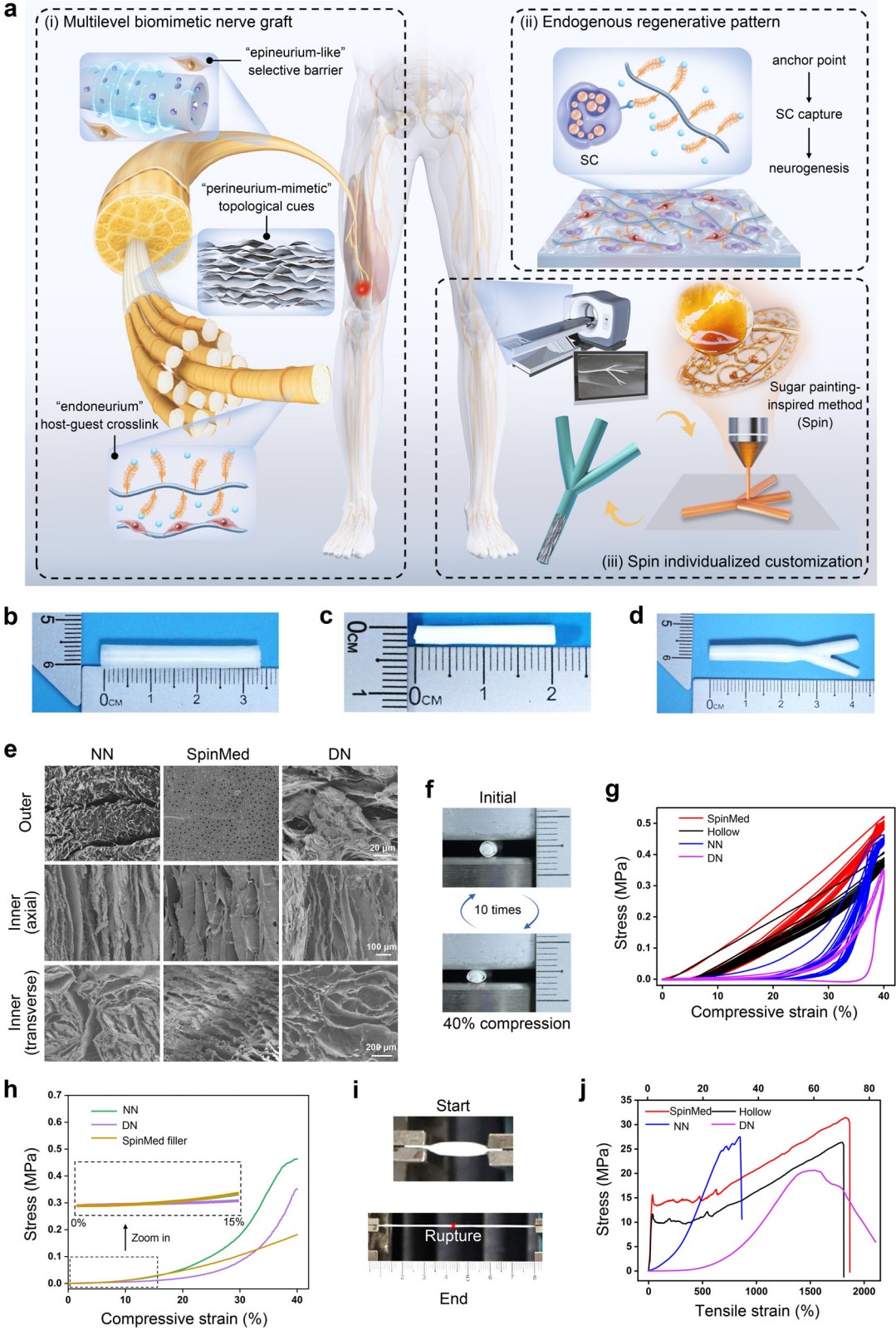

least in part, contributed to distinctive biodegradation behavior and cell adhesion affinities (Supplementary Fig. 3b–e). The function-enhanced epineurium-like sheath of the SpinMed grafts was dramatically superior to the natural structure, exhibiting high permeability to nanoscale proteins and lower permeability to muscle-derived pericytes with micrometer-scale dimensions (Supplementary Fig. 3f–i). These findings revealed that the optimized regeneration cues

outperformed natural morphology simulation strategies, indicating that graft adaptivity to the regenerative microenvironment is a priority consideration in material development and preparation. On the other hand, the fibroin-based SpinMed filler was fabricated through freeze-drying with careful adjustments of the hyaluronic acid content to achieve the optimal filler architecture and rheological properties (Supplementary Fig. 4). Furthermore, we could precisely modulate the

**Fig. 1 | Design and performance of the SpinMed graft. a** Schematic illustration of the design ideas, features, and applications of SpinMed. (i) The multilevel biomimetic strategy for mimicking the epineurium/perineurium to offer structural support and a biomechanical microenvironment and inducing the formation of an endoneurium-mimetic biological unit. (ii) The endoneurium-like regenerative unit is composed of fibroin and endothelial-derived molecules, which bind to SCs. (iii) Sugar painting-inspired additive manufacturing following digital imaging used for individualized nerve defect repair. **b** Gross image of the epineurium-like sheath obtained by the phase separation reverse-mold method after three-dimensional sugar printing. **c** Gross image of the SpinMed graft formed by filler perfusion of the outer shell. **d** Irregular SpinMed fabrication following MRI detection. **e** Scanning electron microscopy (SEM) images of natural nerve, SpinMed, and decellularized nerve with views of the outer surface, longitudinal section, and cross-section. **f** Illustration of compression tests, in which various grafts were restored after 40% compression for 10 cycles. **g** The compression stress-strain curves of the hollow construct, SpinMed graft, natural nerve, and decellularized nerve. **h** The compression stress-strain curves of the SpinMed filler, natural nerve, and decellularized nerve in a single compression. **i** Illustration of tensile tests, in which various grafts were stretched to rupture. **j** The tensile stress-strain curves of the hollow construct, SpinMed graft, natural nerve, and decellularized nerve. NN, natural nerve. DN decellularized nerve. The experiments in **e**, **g**, **h**, and **j** were independently repeated three times with similar results.

filler characteristics, such as by producing a regular topology or random morphology, which exhibited identical chemical properties but distinct physical features (Supplementary Fig. 5). The topology mimicking the perineurium showed aligning cues and effects similar to those of natural nerves and decellularized nerve grafts, achieved through a gradient freezing technique. The compression properties of the SpinMed grafts and other counterparts were investigated (Fig. 1f–h; Supplementary Fig. 3j, k). SpinMed displayed flexibility during repeated compressions, which facilitated the mitigation of mechanical disturbances from muscle contraction after implantation in vivo. Notably, the enhanced physical characteristics were primarily attributed to the outer sheath of SpinMed rather than the filler, while the strain-stress features of the filler were comparable to native nerve tissue during 0 to 20% compression. The tensile stress test revealed that the mechanical stress of the SpinMed graft was significantly superior to that of natural nerves and decellularized nerves, ensuring sufficient adaptability after implantation (Fig. 1i, j).

## SpinMed accelerates axial extension of vessels and increases paracrine MMRN1 signaling

To evaluate the angiogenic capacities of SpinMed, we first performed cell proliferation, migration, and tube formation assays using SpinMed filler in vitro. The EdU assay showed that the proliferative rates of human umbilical vein endothelial cells (HUVECs) in the SpinMed or random counterpart groups were slightly decreased compared to those in the tissue culture plate group, but the difference was not statistically significant ($p = 0.1120$) (Supplementary Fig. 6a, b). Scanning electron microscopy (SEM) observations revealed distinct migration patterns of cells on different interfaces. HUVECs adhered to the SpinMed filler exhibited faster migration than those guided by random cues (Fig. 2a, b). Moreover, cells cultured on SpinMed tended to form regular tubes on Matrigel (Fig. 2c, d). We performed in vivo implantation of SpinMed in rodent sciatic nerve defect models including a mouse 5-mm nerve defect model and a rat 15-mm nerve defect model, by which we evaluated the outcomes of axially vascular extension at 2 weeks after injury. The results indicated that vascular extension, along with the axial architecture of the SpinMed outperformed that of random architectures (Fig. 2e, f).

Proteomics was employed to explore the mechanisms of SpinMed treatment in nerve regeneration. Among all differentially expressed genes between SpinMed and random interface culture, multimerin 1 (MMRN1) was identified as a possible contributor to the angiogenic effects of SpinMed (Fig. 2g). Kyoto encyclopedia of genes and genomes (KEGG) analysis of differentially expressed genes indicated the potential involvement of the coagulation cascade and extracellular matrix (ECM)-receptor pathways (Fig. 2h). Western blot analysis identified increased MMRN1 expression in HUVECs cultured on the longitudinal axis of SpinMed filler (Fig. 2i, j). Complete failure of nerve regeneration was associated with misdirection of the blood vessels, and SC cords tended to follow the vascular regrowth direction towards the MMRN1 beads located in adjacent muscle and away from the bridge (Supplementary Figs. 6c–f and 7). In these cases, SCs migrated into the vascularized regions and deviated from their original

directions, leading to the failure of axonal reconstruction. Moreover, MMRN1-blocking antibody application into the regenerative gap also compromised nerve healing in a rat nerve transection model, in which the number of differentiated SCs within the regenerative zone was positively associated with vessel volume (Supplementary Fig. 6g–j). Regarding the MMRN1-mediated molecular interaction, the structure simulated by the docking model revealed increased paracrine MMRN1 signaling facilitated MMRN1-fibroin binding, and this interaction between the SpinMed filler and vessel paracrine factor established a stable anchorage for future endoneurial formation (Fig. 2k). Immunofluorescence assessment showed more MMRN1 bound to the SpinMed topological filler than to the random control when HUVECs were cultured on these scaffolds (Fig. 2l). Collectively, these results demonstrated that the topological cues of the SpinMed filler accelerated vascular extension, and helped to establish potential anchorages through increased paracrine MMRN1 signaling.

## SpinMed-mediated host-guest crosslinking by high binding affinity between MMRN1 or PECAM1 and fibroin

To verify that fibroin used for fabricating the SpinMed filler had a high binding affinity for two major angiogenic proteins, MMRN1 and platelet endothelial cell adhesion molecule 1 (PECAM1), the docking models, intermolecular contact area, hydrogen bond (HB) number, and binding free energy ($\Delta G$) values were dynamically simulated and measured. The molecular dynamics (MD) simulation displayed intermolecular docking models including hydrogen bond and van der Waals force binding sites (Fig. 3a–f). Compared with polycaprolactone (PCL) and collagen I, fibroin showed additional HB interactions, which was in agreement with its significantly higher affinity for MMRN1. Meanwhile, fibroin exhibited additional hydrophobic interactions with one or more hydrophobic residues of PECAM1 in comparison with PCL and collagen I and thus showed a significantly higher affinity for PECAM1. The $\Delta G$ determined by MD simulation also showed that the fibroin used in the SpinMed filler displayed a higher affinity for the two major angiogenic proteins (MMRN1 and PECAM1) than other commonly used materials (PCL and collagen I) (Supplementary Fig. 8; Supplementary Table 2). Quantification of the intermolecular contact area and HB number showed that fibroin was dramatically superior to PCL and collagen I during 0 to 100 ns, when, especially from 80 to 100 ns, the number of HBs between PECAM1 and fibroin was greater than that between PCL or collagen I and PECAM1 (Fig. 3g–j). These results indicated that fibroin would act as an anchor by binding the regenerative vascular tissue with a higher affinity.

## MMRN1 anchors contribute to the "endoneurium" for angiogenic-neurogenic coupling

To identify the processes of regenerative vessel-mediated neurogenesis and achieve direct angiogenic-neurogenic coupling, we performed a series of experiments, including proliferation, migration, and differentiation detection assays, in a coculture system of vascular endothelial cells and SCs (Fig. 4a; Supplementary Fig. 9a). The EdU assay showed an enhanced proliferative capacity for mouse primary SCs (mSCs) by topological induction but not for RSC96 cells

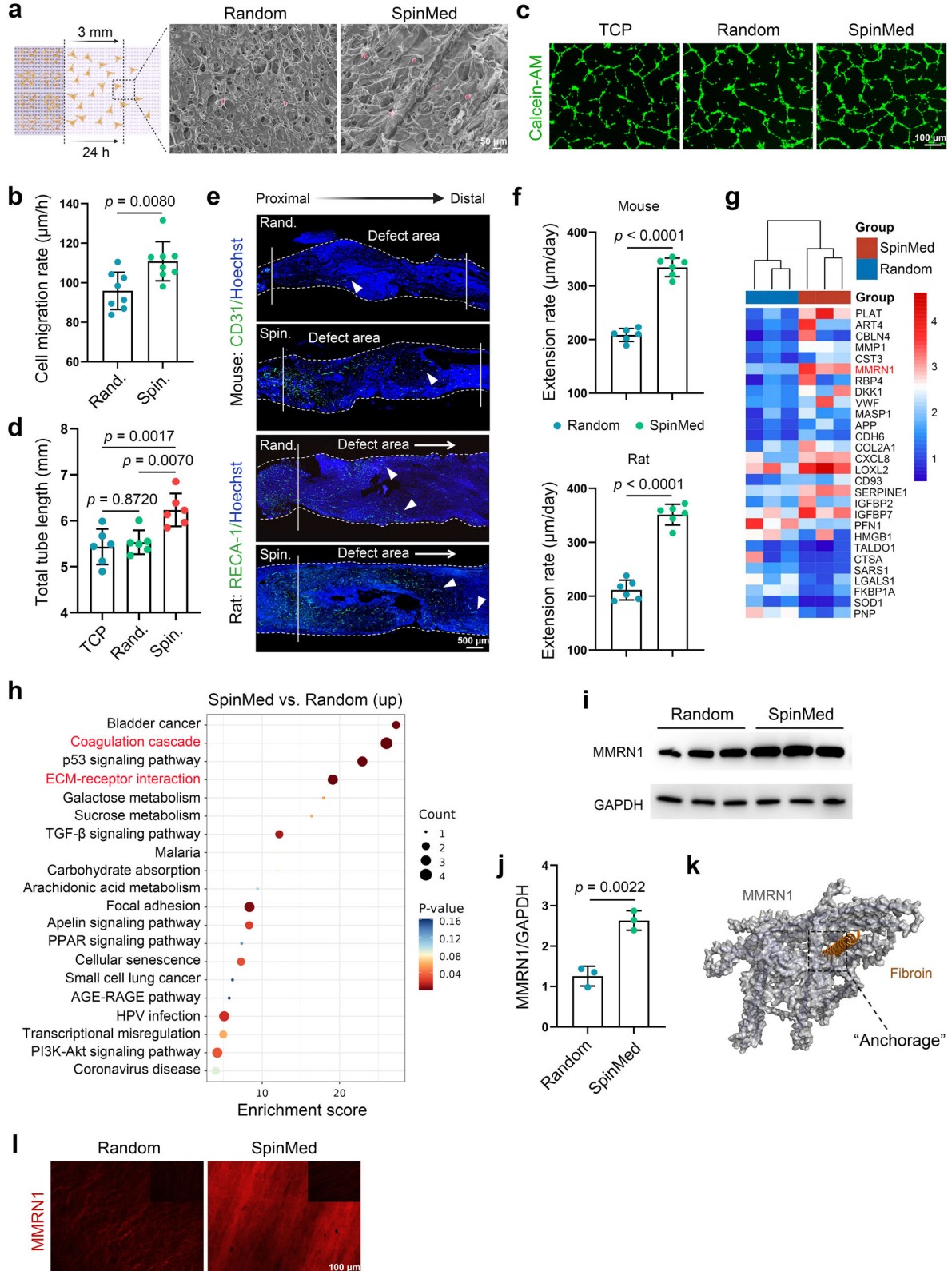

(Supplementary Fig. 9b, c), and the migration rates of SCs determined by transwell assay were accelerated in topology-mediated vascular endothelial cell and SC coculture for both mSCs and RSC96 cells (Supplementary Fig. 9d, e). SC homing and differentiation are necessary for remyelination. Vascular endothelial cells grown upon topological cues exerted significant effects on SC differentiation, as revealed

by the branching SC ratio and branching length of SCs. Serum application was used as a positive control (Fig. 4b, c).

Immunoprecipitation mass spectrometry (IP-MS) and proteomics were cooperatively utilized to investigate the receptors of MMRN1 and the response elements located on SCs. We found that hyaluronan mediated motility receptor (HMMR) and cyclin-dependent kinase 1

**Fig. 2 | SpinMed accelerates axial extension of vessels and increases paracrine MMRN1 signaling. a** Detection of cell migration capacities after 24 h of culture on SpinMed filler or a random counterpart. Panel **a** created with BioRender.com released under CC BY-NC-ND. **b** Quantification of cells that migrated up to 3 mm under SEM ($n = 8$). **c, d** Representative images from the tube formation assay of HUVECs (**c**) and quantification of total tube length after culture with different interfaces (**d**) ($n = 6$). **e, f** Representative images of sciatic nerves harvested from mice or rats 2 weeks after implantation (white full lines represent the initial stump terminal, and arrows indicate extended vessels) (**e**) followed by quantification of vascular extension rates (**f**) ($n = 6$). **g** Heatmap of differentially expressed genes among the SpinMed and random groups ($n = 3$). **h** KEGG analysis of differentially expressed genes between SpinMed and random interface cultures by Limma package. **i, j** Western blot images of MMRN1 expression in vascular endothelial cells (**i**) and quantification of the MMRN1/GAPDH ratio ($n = 3$) (**j**). **k** Simulated model of the MMRN1 protein and fibroin docking site, also considered an anchorage site. **l** Representative immunofluorescence images showing MMRN1 binding to random or SpinMed filler when cultured with HUVECs; the control images presented in the top right corner display the same without cultured cells. TCP tissue culture plate, Rand. random counterpart, Spin. SpinMed. Mean values are shown and error bars represent ± s.d., as analyzed by two-sided Student's $t$-test in **b**, **f** and **j**, or one-way ANOVA with Tukey's post hoc tests in (**d**). The experiments in **e**, **k**, and **l** were independently repeated at least three times with similar results.

(CDK1) were predicted to be the receptors of MMRN1 through direct interactions (Fig. 4d; Supplementary Fig. 9f, g; Supplementary Fig. 10). Subsequently, the intermolecular docking model was examined to illustrate their interactions (Fig. 4e; Supplementary Fig. 10). Several tissue or cell types were submitted to detect HMMR and CDK1 expression, high levels of which were predominant in SCs among potential responsive tissues of peripheral nerves (Supplementary Fig. 9h). We further employed the co-IP assay to confirm the interactions among MMRN1, HMMR and CDK1 in the HEK293T cell line, a transformed human embryonic kidney cell line, verifying the ligand-receptor relationships between MMRN1 and HMMR, MMRN1 and CDK1, as well as HMMR and CDK1 (Fig. 4f). In addition, we analyzed the involved candidate signal transduction pathways by gene set enrichment analysis (GSEA), and found that the coagulation cascade, ECM receptor interaction and oxidative phosphorylation primarily participated in SC activities associated with remyelination (Supplementary Fig. 9i), and this effect was likely to be regulated by the activation of the focal adhesion kinase (FAK) and extracellular-signal regulated kinase (Erk) that FAK/Erk1/2-MAPK signaling pathway (Supplementary Fig. 9j, k). We further evaluated the neurogenic effects driven by conditioned media (CM) collected from SC cultures. CM derived from the SpinMed group altered the axonal length of dorsal root ganglion (DRG) neurons (Fig. 4g, h). Therefore, the MMRN1/HMMR/CDK1 complex drove SC remyelination during SpinMed-mediated angiogenic-neurogenic coupling.

## Accelerated nerve structural and functional restoration by SpinMed in rats

The biosafety of SpinMed was first evaluated after implantation into rats, and the results showed that there was no significant cell toxicity or organ injury (Supplementary Fig. 11). During the period of 12-week follow-up, we performed gait analysis for locomotor activity examination at 8 and 12 weeks after implantation to evaluate the in vivo therapeutic effects of SpinMed compared with other grafts. The denervated paws were prone to shrinking, and the distance between digits was reduced. The rats that underwent SpinMed implantation exhibited improved toe spreading and gait than that from hollow and random groups at both 8 and 12 weeks (Fig. 5a; Supplementary Fig. 12a). The sciatic nerve function index (SFI) was employed to represent the global conditions of motor nerve restoration. Although the SFI evaluated in the SpinMed implantation group did not exceed that in the autograft group, the results showed that SpinMed dramatically restored the motor function of the sciatic nerve comparable to autograft (Fig. 5b; Supplementary Fig. 12b). We further evaluated the recovery of sensory function by assessing responses to thermal stimulation and the presence of mechanical allodynia (Supplementary Fig. 12c). In the hot plate test, rats receiving SpinMed implantation and autograft recovered more quickly from thermal nociception at 12 weeks, showing a reduced withdrawal response time compared with other controls, while no differences in mechanical allodyniano, revealed by Von Frey test, were observed among these groups (Supplementary Fig. 12d, e). We also assessed muscle responses to nerve-derived electrical pulses to reflect the status of the target gastrocnemius muscle. Overall, SpinMed improved electrophysiological recovery compared with its random counterparts and hollow control and achieved therapeutic outcomes similar to those of the "gold standard treatment" autograft at postoperative 12 weeks, as determined by the nerve conduction velocity (NCV) and recovery index of compound muscle action potential amplitude (CMAP) (Fig. 5c–e). These results suggest that SpinMed application is a promising nerve tissue engineering strategy.

Histological assessment was further carried out (Supplementary Fig. 13), along with transmission electron microscopy (TEM) examination of the cross-section as the "gold standard" observation method. We observed the regenerated sciatic nerve morphology by TEM (Fig. 5f) and quantified the myelination and axonal regeneration by the myelin sheath thickness and axonal diameter, and calculated $g$-ratio (Fig. 5g, h). The TEM evaluation revealed the presence of a higher density of myelinated fibers and a thicker myelin sheath within the lumen in the SpinMed group, comparable to those in the autograft group at postoperative 12 weeks. During the regenerative process, we examined the vessel, myelin, and axon at multiple endpoints (Supplementary Fig. 12f), the results showed that enhanced angiogenic activities and vessel volume were observed from SpinMed group samples at postoperative 4 weeks, as well as vessel-myelin-axon regrowth coupling at both 4 and 8 weeks (Supplementary Fig. 12g). Axonal regeneration and myelination were also assessed by immunofluorescence staining for β-tubulin III (Tuj1)/myelin basic protein (MBP) or neurofilament (NF) 200/S100β (Supplementary Fig. 14a–c). SpinMed implantation led to denser nerve fibers and myelinated nerve fibers near the autograft (Supplementary Fig. 14d). Similar results were also found throughout the entire length of regenerated nerves (Supplementary Fig. 14e, f).

Axonal transport drives the signal relay and nutrient supply between the nervous system and the target tissue. FluoroGold (FG) tracer injection followed by histological detection of DRG neurons and spinal cord anterior horn (AH) tissue was employed to detect the transport capacities of regenerative axons (Supplementary Fig. 14g). Autofluorescence images showed that more positive sensory neurons within the DRG and motor neurons within AH regions after SpinMed implantation than those from hollow and random counterpart groups (Supplementary Fig. 14h), and furthermore, quantification analysis supported the comparable effects of the SpinMed graft and autografts, demonstrating favorable neurofiber reconnection and transport function restoration after 12-week SpinMed repair (Supplementary Fig. 14i). These injury signals and component alterations may also result in spinal cord remodeling. Nestin and glial fibrillary acidic protein (GFAP) were labeled to determine the number and distribution of neurons and glial cells within the entire spinal cord (Fig. 5i; Supplementary Fig. 15). Muscles are highly innerved to exert contract function, and long-term denervation leads to muscle atrophy and dysfunction. SpinMed application to a nerve defect mostly maintained the morphology and wet weight of the gastrocnemius examined at 12 weeks, but the effect was slightly inferior to that of autograft application (Fig. 5j, k; Supplementary Fig. 16).

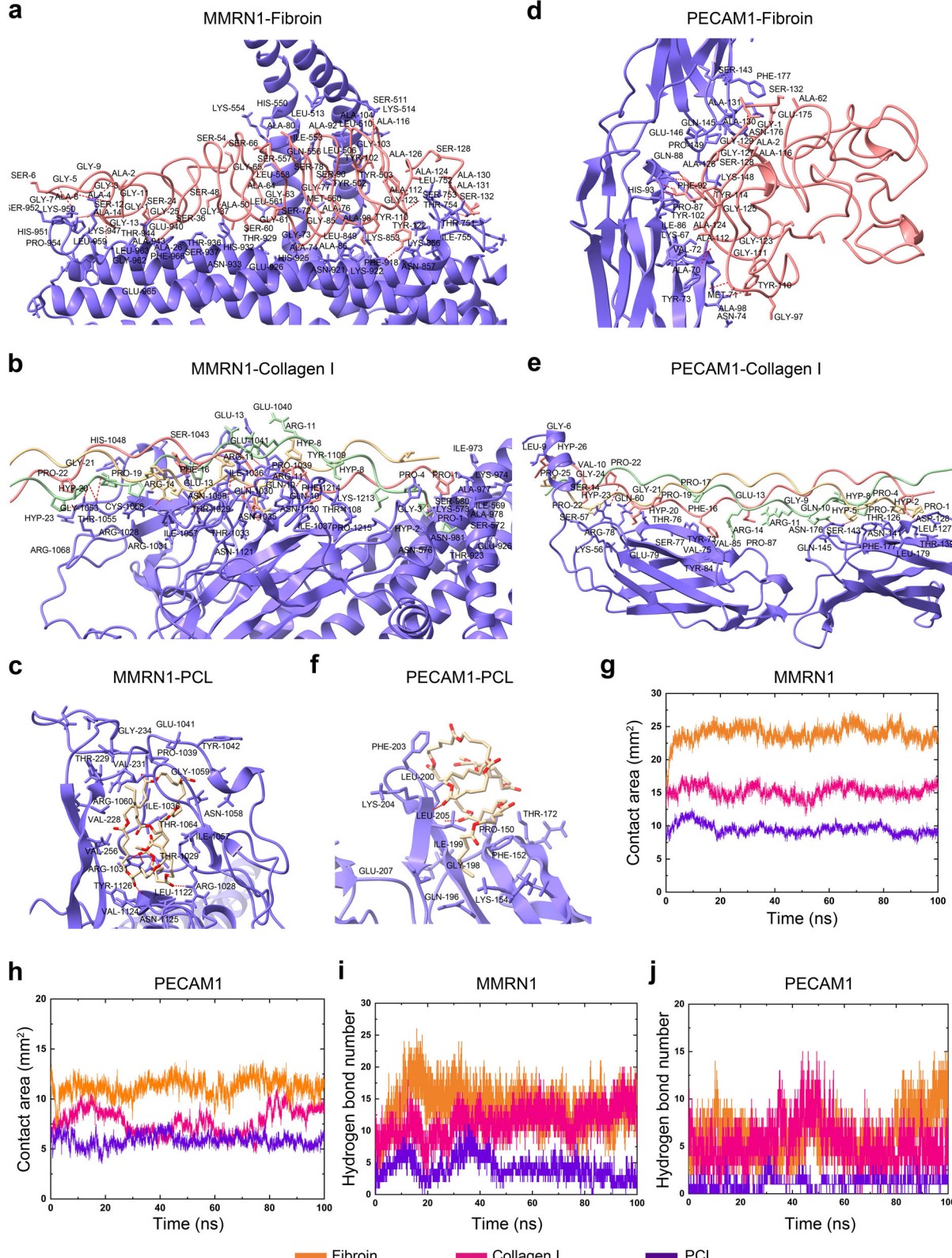

**Fig. 3 | Fibroin within SpinMed filler facilitates MMRN1 and PECAM1 binding.** **a**–**c** Detailed information on MMRN1 binding to fibroin (**a**), collagen I (**b**), and PCL (**c**) at docking sites. Red dashed line: hydrogen bond. **d**–**f** Detailed binding sites of PECAM1 docking to fibroin (**d**), collagen I (**e**) and PCL (**f**). Red dashed line: hydrogen bond. **g**, **h** Contact area of MMRN1 (**g**) or PECAM1 (**h**) with fibroin, collagen I and PCL within 0–100 ns, calculated by molecular dynamics simulation. **i**, **j** The number of hydrogen bonds between MMRN1 (**i**) or PECAM1 (**j**) and fibroin, collagen I and PCL. PCL polycaprolactone, MMRN1 multimerin 1, PECAM1 platelet endothelial cell adhesion molecule 1. The computational simulations were independently repeated at least three times with similar results.

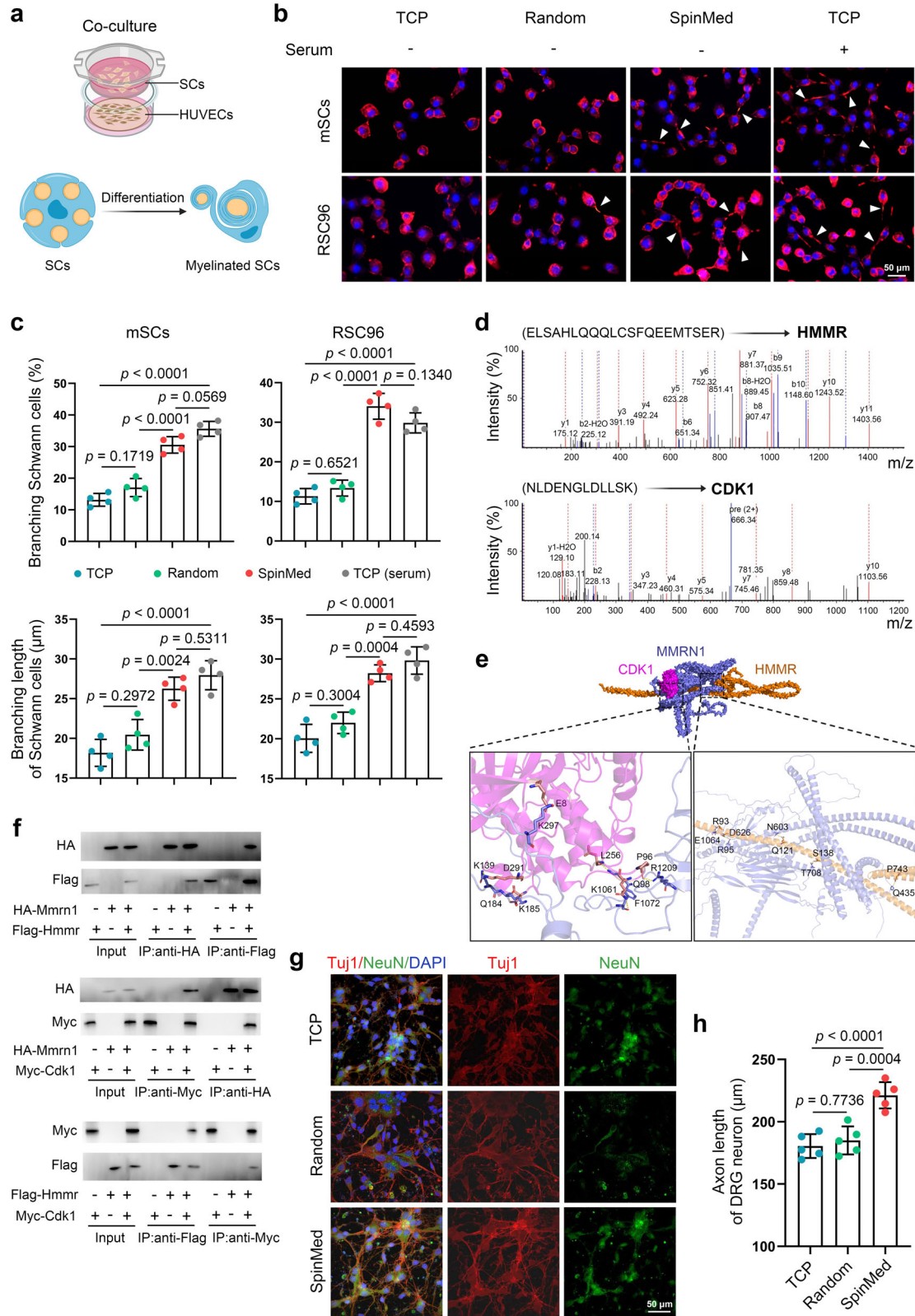

**Complicated nerve defect repair with reliable biosafety via an individualized strategy in canines**

Commercial nerve grafts used in current clinical practice often fail to cure complicated nerve defects due to mismatched size and shape; however, the SpinMed graft designed in this study addresses, at least in part, these issues. The individualized SpinMed graft and other controls were fabricated and implanted into a beagle sciatic defect model for

12 months (Fig. 6a, b; Supplementary Fig. 17a). We monitored the biosafety of SpinMed by detecting the serum levels of aspartate aminotransferase (AST), alanine aminotransferase (ALT), lactate dehydrogenase (LDH) and blood urea nitrogen (BUN) at postoperative 4 weeks, 16 weeks and 12 months, and there was no significant alteration of the serum levels of these molecules (Supplementary Fig. 18). Furthermore, the SpinMed could achieve nearly complete

**Fig. 4 | MMRN1/HMMR/CDK1 complex contributes to SpinMed-induced angiogenic-neurogenic coupling. a** Schematic illustration of the HUVEC and SC coculture system. Panel **a** created with BioRender.com released under CC BY-NC-ND. **b, c** Representative images of SC (mSC or RSC96 cell) differentiation when cocultured with HUVECs on various surfaces (**b**) and quantification of branching cell percentages and average branching lengths (*n* = 4) (**c**). **d** Mass spectrogram of identified HMMR and CDK1 binding to MMRN1 after immunoprecipitation (IP) (*n* = 3). **e** Molecular docking model of MMRN1, HMMR, and CDK1, as well as docking sites by hydrogen bonds. **f** IP analysis of binding among MMRN1, HMMR, and CDK1 using the HEK293T cell line. **g, h** Representative immunofluorescence images of DRG neurons cultured with Schwann cells after various treatments (**g**) and quantification of axon extension lengths (*n* = 5) (**h**). TCP, tissue culture plate. HMMR, hyaluronan mediated motility receptor. CDK1, cyclin-dependent kinase 1. Mean values are shown and error bars represent ± s.d., as analyzed by one-way ANOVA with Tukey's post hoc tests in (**c**) and (**h**). The experiments in **f** were independently repeated three times with similar results.

degradation without obvious organ toxicity as revealed by long-term followup (Supplementary Fig. 18). Magnetic resonance imaging (MRI) detection revealed that the individualized SpinMed grafts restored branched nerve defect continuity well, as indicated by complete structural reconstruction, while grafts with straight structures only germinated nerve trunks but not clear branches (Supplementary Fig. 17b). Electrophysiological assessment showed that CMAP amplitude recovery index was improved by the individualized SpinMed grafts at postoperative 16 weeks and 12 months, indicating that precise structural reconstruction may also facilitate functional restoration (Fig. 6c; Supplementary Fig. 17c). Dynamic motor evaluation of the right hind limb joints of beagles showed greater flexibility in the SpinMed group, as indicated by the ankle joint and metatarsophalangeal joint flexion angle, and locomotor capacity grade (Supplementary Fig. 19; Supplementary Movie 1 to 4).

We selected 4 representative sites for TEM observation and toluidine blue (TB) staining of the transverse ultrastructural view at postoperative 16 weeks and 12 months (Fig. 6d; Supplementary Fig. 17d). At postoperative 16 weeks, the specimens collected from the individualized SpinMed group showed comparable ultrastructure to autografts in the proximal terminal (P1), with some even superior to autografts and the other counterparts at branching sites. The regenerative nerve collected from SpinMed group exhibited better myelin sheath thickness and *g*-ratio within P3 and P4 (Supplementary Fig. 17e, f; Supplementary Fig. 20). Moreover, neural integrity restoration by SpinMed also showed superiority to single counterpart, decellularized graft, and even autograft at postoperative 12 months, as revealed by TEM parameter analyses, in which myelin sheath thickness displayed significant difference (Fig. 6e–g; Supplementary Fig. 21). Histology for regenerative nerve and innervated gastrocnemius muscle was performed to evaluate neural morphology and function. We found the number of myelinated nerve fibers was comparable to autograft, and targeted muscle atrophy was also significantly improved compared with decellularized graft application (Fig. 6h, i; Supplementary Fig. 22). These findings collectively demonstrated the practical value and translational potential of SpinMed.

## Discussion

We conducted an individualized design process without any loading components and synthesized a multilevel neurium-mimetic nerve graft named "SpinMed" composed of modified epineurium, topological perineurium, and spontaneously formed endoneurium to allow the most adaptation for nerve regeneration. The outer structure mimicking the epineurium serves as a selectively physical barrier for limiting access to cicatricial fibrosis following injury, as abnormal tissue invasion has been identified to inhibit neural repair[32,33]. Enhanced mechanical properties conferred by the hollow PCL sheath beyond those of the native counterpart supported stump regrowth. A porous interface (pore size of approximately 1 μm), acquired by phase separation, permitted nanoparticles to cross but rejected cells in the radial direction. This method easily achieved porous architecture placement during solvent evaporation, and notably, this procedure saved a massive time and work compared with previous techniques[34]. Consequently, the enhanced epineurium allowed sufficient space for nerve growth, superior to

allogeneic grafts, and importantly, this modified design based on regenerative preferences inspired us to revolutionize the concept of biomimetic medicine. Aligned filler with a less than 100-μm cue was designed to simulate the perineurium as an efficient adhesive media[35], guiding oriented extension of the nerve bundle in a higher level of regenerative efficiency. The micropattern in the present study exhibited the desired pro-regenerative performance (for example, cell migration and angiogenesis), superior to that in previous studies[10,36]. These findings suggest that this biomimetic strategy may also be effective for other tissue types, such as blood vessels, tendons, and bone. Future studies will determine the distinguished topological features of scaffolds as supporting and instructive structures for matching various tissue preferences.

It is well established that excessively small topological cues (smaller than a single cell) often fail to induce cell adhesion, survival, and desired behavior[37,38]. In the biomimetic design of the delicate endoneurium, it is difficult to load specific cues into nerve grafts using classical tissue engineering strategies. Importantly, reconstruction of vessels within the endoneurium is a prerequisite for neural tissue regrowth and potentially helps to trigger remyelination[39–41]. A rRelevant study indicated that silk fibroin with suitable surface structure and mechanics may activate the direct conversion of cell fate to a phenotype that facilitates tissue repair[42]. Therefore, in this study, we promoted endogenous biological induction of endoneurial self-organization by placing silk fibroin, identified to exhibit stronger adhesion to angiogenic factors than traditional NGC materials (e.g. PCL and collagen) onto a perineurium-like filler[43]. As identified by in vitro and in vivo studies, the SpinMed graft in the present study successfully accelerated vascular extension. Both computational and experimental approaches confirmed that fibroin as an anchor closely binds to the angiogenic marker PECAM1 and vessel-nerve coupling protein MMRN1 via chemical hydrogen bonds, more than PCL and collagen type I. The binding complex as a functional unit mainly contributed to bioactive endoneurial construction and directly guided Schwann cell homing, differentiation, and remyelination, as well as axon extension through MMRN1/HMMR/CDK1-mediated vessel-nerve coupling. These findings suggest a reliable concept for biomaterial-inspired neural organogenesis for the construction of peripheral nerve organoids.

Sugar painting, as traditional Chinese art, provides inspiration for the development of tissue engineering strategies, and we have previously employed it to fabricate a heart repair device, successfully facilitating heart tissue regeneration[44]. The flexible control and individualized parameter determination of this economical method with reliable biosafety meet the requirements of ideal nerve graft customization, especially in cases of complicated structural damage, overcoming the challenges of current nerve tissue engineering methods[18,45]. Notably, well-fabricated nerve grafts currently used in clinical practice fail to cure defects with complicated shapes and extremely unpredictable lengths; therefore, the shape and size of nerve grafts are vital for efficient nerve repair[14–16]. In the present study, rapid and individualized fabrication based on anatomy and digital imaging, combined with phase separation and gradient freeze-drying, enables regenerative nerves to extend with axial guidance, alleviating the treatment dilemma. This suggests that accurate digital imaging-

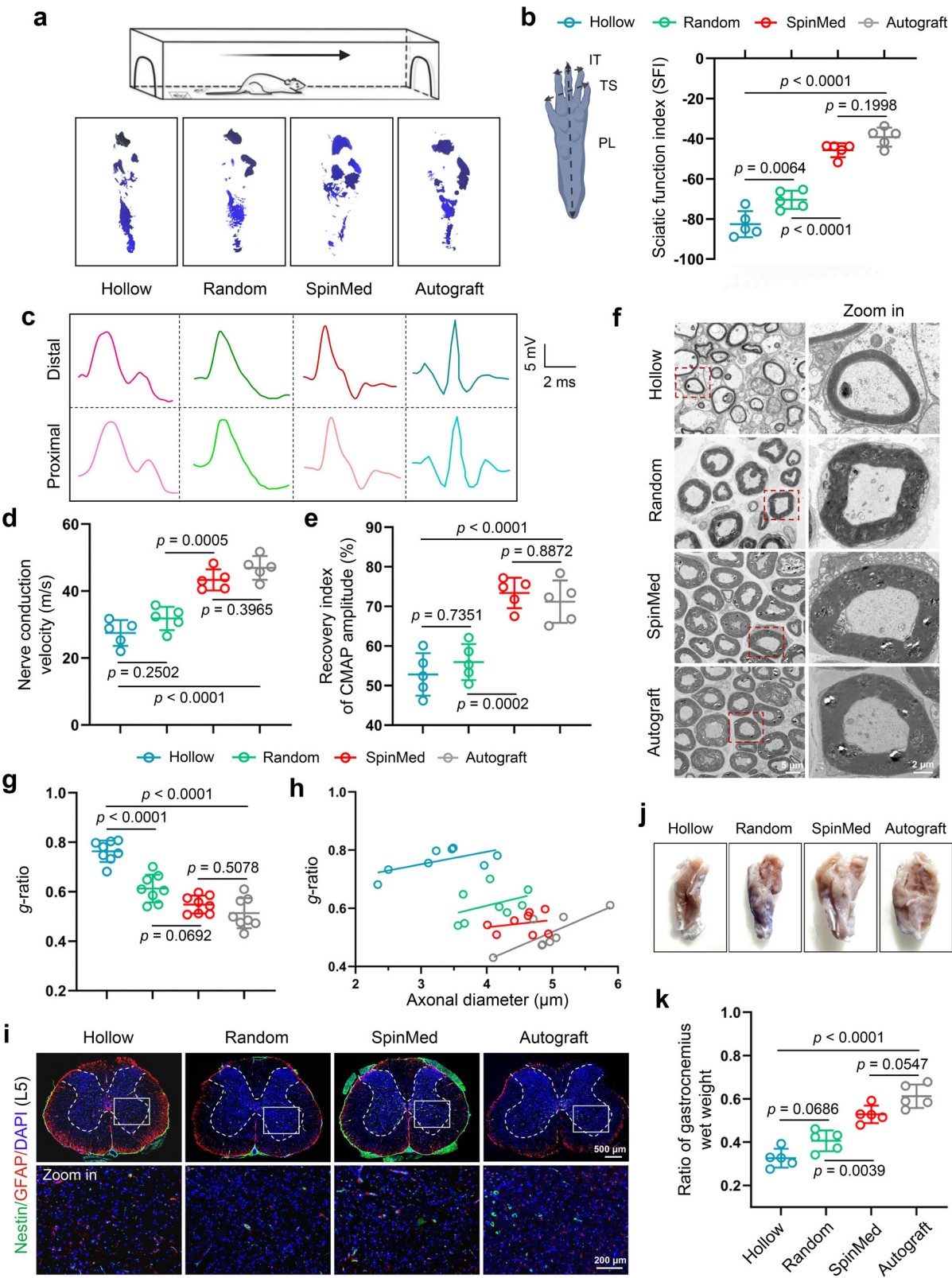

guided strategy formulation, especially in tissue engineering, will potentially facilitate precision medicine.

Another appealing translational feature of SpinMed is that most components, including PCL and silk fibroin, are Food and Drug Administration (FDA)-approved and favorable for clinical translation. They exhibit excellent biocompatibility, unique mechanics, and optimized biodegradability, all contributing to the production of a satisfactory nerve graft for preclinical assessment, as revealed by up to 12-month follow-up in rodents and canines. The slow degradation rate of PCL allows it to provide long-term physical support for meeting the needs of electric signal transduction restoration, whereas both fibroin and decellularized nerve matrix as functional filler display early- to medium-term degradation. The latter makes it possible for the adaptive degradation rate to be

**Fig. 5 | SpinMed restores peripheral nerve morphology and function in rodents. a, b** Schematic illustration of gait evaluation (**a**, top panel), representative images of footprints (**a**, bottom panel), and quantification of sciatic function index (*n* = 5) (**b**). Panel **a** and **b** were created with BioRender.com and released under CC BY-NC-ND. **c–e** Representative electromyographic patterns detected in each group (*n* = 5) (**c**) and quantitative analysis of nerve conduction velocity (**d**) and CMAP amplitude recovery index (**e**). **f–h** Representative TEM images of regenerative nerve tissue (transverse view, left panel) and zoom-in images (right panel) from various groups (*n* = 8) (**f**). **g, h** Quantification of parameters for the nerve ultrastructural analysis, including the axon diameter and myelin sheath thickness, shown by the *g*-ratio (**g**), and linear regression of the *g*-ratio and axon diameter (*n* = 8) (**h**). **i** Representative immunofluorescence images of Nestin and GFAP expression in the spinal cord (L5) (*n* = 5). **j, k** Gross view of gastrocnemius harvested from various groups (*n* = 5) (**j**) and quantification of gastrocnemius wet weight compared to contralateral counterpart (**k**). CMAP, compound muscle action potential. Mean values are shown and error bars represent ± s.d., as analyzed by one-way ANOVA with Tukey's post hoc tests in **b**, **d**, **e**, **g**, **h** and **k**.

modulated by host-guest crosslinking and provide sufficient space and appropriate direction for neural repair; furthermore, the degradation products (e.g. amino acids and polypeptides) are expected to furnish newly regenerative tissue[46,47]. These biocompatible molecules show promising therapeutic outcomes in peripheral nerve regeneration, as well as biosafety, supporting their clinical application.

In the past decade, naturally derived extracellular matrix from allogeneic sources (Avance® Nerve Graft (Axogen) and decellularized graft used in the present study) and collagen-type-I-based NGC (NeuraGen® Nerve Guide (Integra)) have dominated in the field of neural repair[48,49]. Limited donor sources, size mismatching, and lack of mass production are major factors restricting their clinical usage. Here, we demonstrate the possibility of the broad application of SpinMed through preclinical assessments, supporting its relevance in the treatment of numerous nerve injury types. The superiority of SpinMed in a nonhuman primate model is under investigation, and a clinical trial will be conducted to evaluate its clinical translation. Overall, SpinMed is capable of a wide range of applications, such as in the management of massive defects, irregular injuries, and multi-branched damage. The promising concept and design may also revolutionize the fields of vessel, tendon, or bone regeneration.

## Methods

### Materials and reagents
Hexafluoroisopropanol (HFIP) was purchased from Adamas, China. PCL was purchased from Sigma-Aldrich, USA (Mn 80,000 g/mol). Sugar was purchased from Shanghai Sugar Cigarette and Wine Group, China (source). Silk fibroin (SF; 8-10 kDa) was purchased from Engineering For Life, China. Hyaluronic acid (HA; 97%) was purchased from Macklin, China. Phosphate-buffered saline (PBS) was purchased from Sigma-Aldrich, USA. All reagents were used as received without further purification unless otherwise noted.

### Human subjects
We acquired 23 nerve tissue specimens from 23 independent patients (16 male and 7 female individuals, age ranging from 31 to 56 years) who had suffered amputation due to posttraumatic complications. The human specimens used in this study were obtained from clinical practices, which were approved by the Ethics Committee of Shanghai Sixth People's Hospital (no. 2022-KY-200(K)). Written informed consent was obtained from all patients.

### Generation of αSMA-tk mice
A truncated version of the herpes simplex 1 virus thymidine kinase (HSV1-tk) gene with 3'UTR was first amplified. The extended αSMA promoter was cloned and inserted into the pCR2.1-TOPO vector (Invitrogen) using EcoRI and NotI restriction sites, and then this αSMA promoter-TOPO construct and the tk fragment were digested and ligated together. The whole αSMA-tk construct was released from the vector using EcoRI and XbaI before purification and injection into fertilized eggs. The transgenic mice were generated on an FVB background and backcrossed for at least 10 generations onto a BALB/c genetic background.

### Fabrication of SpinMed
The hollow (outer sheath) of SpinMed was fabricated using a 3D printing technique combined with a phase separation reverse-mold method, as described in our previous work[44]. First, we constructed a neural damage model with detailed parameters using digital imaging detection and performed sugar-based individualized template remodeling using 3D printing (GeSim BS 4.2). Sugar was preheated for 1 h at 160 °C in heated storage to remove any residual water and then printed in various shapes at 130–145 °C. Printing speeds were adjusted to keep the ink flowing steadily and create smooth templates. The templates were immersed in the PCL solution and left to evaporate in a ventilated area six times. Then, the caramel templates covered with multiple layers of PCL were soaked in distilled water for 24 h to remove the caramel-based template, followed by freeze-drying in a vacuum.

We further fabricated the biomimetic perineurial filler based on the hollow construct prepared above. The filler was composed of decellularized neural extracellular matrix (dnECM), HA, and water-soluble SF. They were dissolved in lab-made deionized water (5%, wt/vol) at a 1:2:4 weight ratio. The mixed solution was drawn by a syringe and injected into the hollow construct to ensure that all parts were fully filled with the solution and then transferred to 6 °C for 2 h, 0 °C for 15 min, and −190 °C until the ice crystals grew completely along the axial direction of conduits, followed by freeze-drying in a vacuum.

### Mechanical properties
The mechanical properties of the SpinMed graft were evaluated using a universal testing machine equipped with TestSuite TW software and a 100 N load cell. All procedures were performed at room temperature, and at least three specimens were tested and averaged for each group. In all tensile and compression tests, the candidates retained their cylindrical shapes with a length of 15 mm. For the uniaxial tensile tests, the deformation rate was maintained at a rate of 50 mm min⁻¹ until sample rupture, and the compressive tests at a rate of 1 mm min⁻¹. During compression tests, each sample was compressed to a strain of 40% and then allowed to recover up to 5% before immediately being compressed to 40% again for 10 cycles.

### Contact angle test
The contact angles of the SpinMed outer shell and filler were examined over time with an automatic video micro contact angle-measuring instrument (OCA 40 Micro, DataPhysics). The filler solution and the PCL in HFIP solution were respectively poured into teflon molds (60*60*20 mm) and freeze-dried to a film. The distilled water fell on the surfaces of samples with a volume of 5 μl and a velocity of 1 μl s⁻¹. Three independent samples in each group were measured and repeated for three times at different time points (0, 1, 2, 3, 5, 10, 15, 20, 25, 30 s) to plot the contact angle curves.

### SEM detection and FTIR examination
The microstructures of SpinMed and counterparts were examined from various views using an SEM device (3.0 kV, SU8010, Hitachi), and the images of outer surfaces, cross-section, and axial section were captured. The infrared absorption spectra of the SpinMed filler before

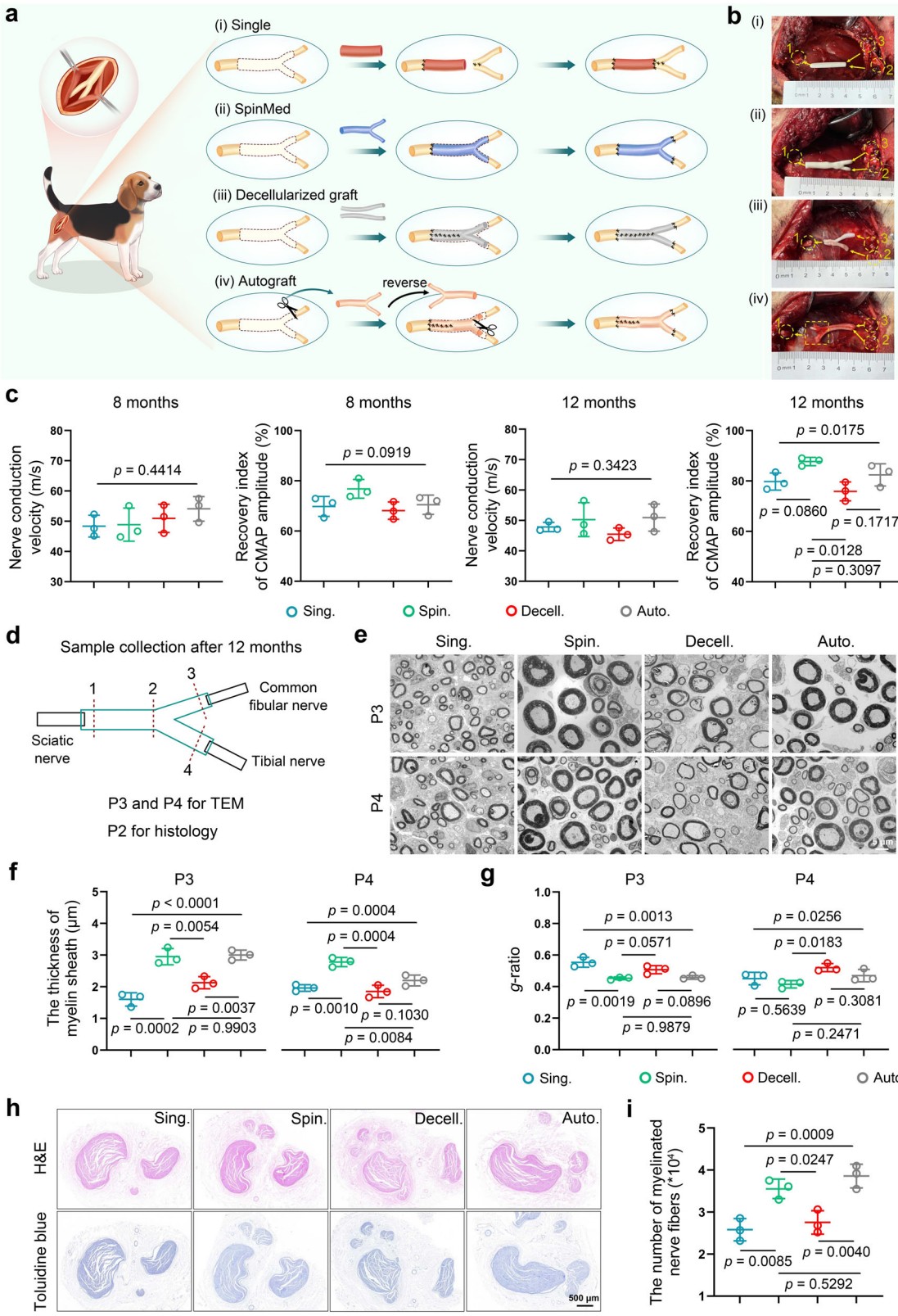

and after dopamine modification and IKVAV immobilization were detected by an FTIR (Nicolet iS50) spectrometer in reflectance mode. For each spectrum obtained, the sample was placed into the measurement chamber, and 64 scans in total were accumulated with a resolution of 4 cm⁻¹. Scanning was conducted in the wavenumber range from 400 to 4000 cm⁻¹.

## Cell culture

HUVECs were purchased from Procell Co., Ltd. in Wuhan, China. Cell identification was performed by immunostaining with PECAM1. The HUVECs were maintained in Endothelial Cell Growth Medium (ECM, ScienCell) supplemented with 10% fetal bovine serum (FBS), 1% penicillin/streptomycin (P/S), 1% endothelial growth supplements, and

**Fig. 6 | SpinMed, as an individualized preclinical product repairs complicated nerve defects in canines with long-term followup. a** Illustration of surgical procedures performed in complicated sciatic nerve defects in the "Single" "SpinMed" "Decellularized graft" and "Autograft" groups. **b** Surgical images of application methods in various groups, with labels for sciatic nerve (1), tibial nerve (2), and common peroneal nerve (3). **c** Analysis of nerve conduction velocity and CMAP amplitude recovery index by electrophysiology assessment at postoperative 8 and 12 months ($n = 3$). **d, e** Illustration of the sample collection positions at postoperative 12 months for TEM detection (**d**), where transverse ultrastructural views were obtained ($n = 3$) (**e**). Panel **d** created with BioRender.com released under CC BY-NC-ND. **f, g** Quantification of parameters for the nerve ultrastructural analysis at the P3 and P4, including the myelin sheath thickness (**f**) and the $g$-ratio (**g**) ($n = 3$). **h, i** Representative images of H&E (upper row) and toluidine blue (lower row) staining for regenerative nerves collected from the P2 at postoperative 12 months ($n = 3$) (**h**), and quantification of the number of myelinated nerve fibers (**i**). Sing. single counterpart, Spin. SpinMed, Decell. decellularized graft, Auto. autograft. Mean values are shown and error bars represent ±s.d., as analyzed by one-way ANOVA with Tukey's post hoc tests in (**c, f, g,** and **i**).

cultured at 37 °C, 5% $CO_2$ in a humidified environment. The medium was replaced every other day, and HUVECs were passaged at 80–90% confluence. The cells at passage two were used in the following experiments.

The primary mSCs and rat RSC96 cells were purchased from the American Type Culture Collection (ATCC). They were cultured in Dulbecco's modified Eagle medium (DMEM; Gibco) supplemented with 10% FBS and 1% P/S in a 5% $CO_2$ incubator at 37 °C.

## HUVECs tube formation

To perform the tube formation assay, 24-well plates were pre-coated with 200 μL Matrigel (Corning) for 30 min at 37 °C. HUVECs were then seeded into the plates at a density of $1 \times 10^5$ cells per well and cultured with serum-free media for 6 h. The living HUVECs were stained with Calcein-AM (2 μg/ml) (Sigma-Aldrich) for 30 min at 37 °C and 5% $CO_2$. After the replacement of the media containing Calcein-AM with serum-free media, images displaying capillary-like structures were captured by a fluorescence microscope (Leica DMi8). The capacity of tube formation was quantified by calculating the total length of tubes and branch points per field using Image J software.

## Cell proliferation assay

Cell proliferation was detected using a cell counting kit-8 (CCK-8; Dojindo) according to the manufacturer's instructions. Aliquots containing $5 \times 10^3$ cells were seeded into 96-well plates and cultured for 24 and 48 h. At the detection time points, cells were incubated with CCK-8 for a further 2 h, after which the optical density (OD) values were measured at 450 nm using a microplate reader (BioTek).

Cell proliferation was also examined by an EdU detection kit (Ribo) according to the manufacturer's protocol. Briefly, cells were seeded onto 96-well plates and cultured for 48 h. At the detection time points, 0.5 μl of 50 μm EdU solution was added to each well containing 500 μl of medium for 2 h. The cells were fixed using 4% paraformaldehyde (PFA; Solarbio) and incubated with 2 mg/ml glycine solution for 5 min with oscillation. The cells were incubated with 100 μl of the penetrant to each well with 10 min of oscillation followed by 100 μl of Apollo staining solution for 30 min. After rinsing with methanol, Hoechst 33342 was used to stain cell nuclei. Images were observed under a fluorescence microscope (Leica DMi8).

## Molecular dynamics simulation

MD simulations of fibroin, collagen type I and PCL with MMRN1 and PECAM1 were respectively performed by using GROMACS 2021.5 package. Proteins were parameterized by Amberff14sb force field. PCL ($n = 10$) was geometrically optimized by Gaussian 16 under an implicit water solvation model with density functional theory B3LYP/def2-SVP level and DFT-D3 dispersion correction. Ambertools21 and ACPYPE were used to construct the general AMBER force field 2 (GAFF2) parameters, and Multiwfn was used to fit the restrained electrostatic potential 2 (RESP2) charge. The potential binding models of fibroin, collagen type I, and PCL on MMRN1 or PECAM1 were achieved by HDOCK and AutoDock Vina 1.2.3, then the best docking models were used for further MD simulations, and 3D models were visualized by UCSF ChimeraX.

Cubic box was established and solvated in TIP3P water, then 0.15 M NaCl was added to keep it electrically neutral. Energy minimization was carried out using the steepest descent algorithm with a force tolerance of 500 kJ/mol nm. These systems were relaxed for 1 ns under NPT ensemble, and position restraints with a constant of 1000 kJ/mol nm in three directions were performed on heavy atoms of protein and nucleic acid. Following it, a 100 ns NPT MD simulation was performed. Pressure was maintained at 1 bar by the Parrinello-Rahman barostat in an isotropic manner, and the temperature was maintained at 310 K by the V-rescal thermostat. The LINCS algorithm was performed to constrain the bond lengths of hydrogen atoms. Lennard-Jones interactions were calculated within a cutoff of 1.2 nm, and electrostatic interactions beyond 1.2 nm were treated with particle-mesh Ewald (PME) method with a grid spacing of 0.16 nm. Based on these, the binding free energies of fibroin, collagen type I, and PCL with MMRN1 and PECAM1 were analyzed according to the MM/PBSA method by the gmx_MMPBSA tool[50].

## SC migration detection

Cell migration capacity was evaluated using 8.0 μm pore size transwell chambers (Corning). SCs at a density of $2 \times 10^4$ cells in 200 μl basal medium with 1% FBS were loaded into the upper chamber, which was inserted into a 24-well plate with 500 μL of complete medium in the lower chamber. After 24 h, cells that migrated across the transwell member were fixed with 4% PFA and stained with 0.1% crystal violet (Solarbio). The migration activity was quantified by counting the number of migrated cells under a light microscope (Nikon TE2000-E).

## SC differentiation assay

mSCs or RSC96 cells were seeded on the confocal dish (NEST) at a density of $5 \times 10^3$ cells/dish with serum-free medium. After cell attachment, the culture medium of SCs was replaced with the conditioned medium of HUVECs (rewarmed to 37 °C) to induce SCs differentiation, and the medium with 10% FBS or without FBS served as positive and negative controls, respectively. After 24 h, the cells were fixed with 4% PFA and stained for F-actin fibers and nuclei using phalloidin-TRITC (Invitrogen) and 4,6-diamino-2-phenyl indole (DAPI; Sigma-Aldrich). Images were observed under a fluorescence microscope (Leica DM6).

## DRG neurites extension detection

DRGs with redundant roots were dissected from postnatal 10-day SD rats and collected in a cold neurobasal medium. The DRGs were incubated in collagenase at 37 °C for 30 min and rotated for 30–35 min at room temperature. Then, cells were centrifuged at 1000 g for 2 min. Supernatants were removed, and pellets resuspended with 1 mL adult DRG culture medium by trituration with 200-μL pipette tips 10 times. Cell suspensions were transferred to new tubes through a cell strainer and spun at $200 \times g$ for 5 min. The cells were resuspended in neurobasal media containing 2 mM glutamine, 2% B27 supplement, and 1% penicillin/streptomycin. Cells in suspension were plated on 8-mm glass coverslips. The culture medium was then changed to 50% neurobasal media (Gibco) plus 50% SCs CM from different substrates. After 3 days, the cells were fixed with 4% PFA and stained for neuron neurites, SCs

and nuclei using anti-beta III Tubulin (Tuj 1) and anti-NeuN and DAPI, respectively. The confocal laser scanning microscope (CLSM; Leica TCS SP5) was used for imaging.

## Proteomic analysis

The total protein of CM or SCs was collected for proteomic (DIA) detection. The protein was sequentially digested with trypsin at a ratio of 1:40 (enzyme to substrate) for 16 h at 33 °C. The tryptic peptides were then acidified with 1% trifluoroacetic acid (TFA, pH 2–3) prior to C18 desalting with Sep-Pak Vac (1 cm$^3$, 50 mg) C18 cartridges (Waters, MA, USA) according to the manufacturer's protocol. Desalted peptides were then dried under vacuum and dissolved in 20 μL of MS buffer containing 0.1% formic acid and 2% acetonitrile in water (HPLC grade). The peptide concentration was measured by Nanoscan (Analytik Jena AG) at an absorbance of 280 nm. Peptides (0.4 μg) were separated over 30 min on an LC gradient using an analytical column and added to a mass spectrometer. The resolution of MS1 was 60,000, and that of MS2 was 30,000. OpenSWATH (version 2.0.0 Sep 26 2017) was performed against public libraries. Pyprophet limited the peptide and protein identification to a 1% false discovery rate (FDR). GO term and KEGG pathway enrichment analysis were used for the biological categorization of the significantly differentially expressed proteins.

## Western blotting

Protein was extracted from tissue or cells using a radio-immunoprecipitation assay (RIPA) lysis buffer (Sigma-Aldrich). The concentrations of protein samples were determined using a bicinchoninic acid kit (Thermo Fisher). Protein was separated by 8%-12% sodium dodecyl sulfate-polyacrylamide gel electrophoresis (SDS-PAGE; Bio-Rad) and then transferred onto a polyvinylidene difluoride membrane (PVDF; Millipore). The membrane was blocked with 5% (w/v) BSA at room temperature for 1 h and incubated with primary antibodies at 4 °C overnight followed by incubation with secondary antibodies at room temperature for 1 h. After rinsing with PBST three times, the candidate proteins were visualized using Omni-EC™ Femto Light Chemiluminescence Kit (Epizyme), in which unprocessed images were provided in this paper (Supplementary Fig. 23).

## Immunoprecipitation

Immunoprecipitation was performed using the HEK293T cell line transfected with the HA-Mmrn1 plasmid, Flag-Hmmr plasmid, or Myc-Cdk1 plasmid. Then, the cell membrane lysate was collected and immunoprecipitated by incubation with antibodies against HA, Flag, and Myc, respectively, followed by adsorption to protein G Sepharose. Immunoprecipitants were separated by SDS-PAGE and transferred onto PVDF membranes. The membrane was incubated with antibodies against HA, Flag, and Myc, and visualized by using an Omni-EC™ Femto Light Chemiluminescence Kit.

## Mass spectrometry

The membrane protein of the HEK293T cell line was extracted using the Membrane and Cytosol Protein Extraction Kit (Beyotime). The antibody was incubated with protein lysate supernatant for 12 h at 4 °C. Protein A/G magnetic beads (Epizyme) were incubated with lysis/wash buffer for 30 min at room temperature three times and then combined with the antibody-antigen complex. After washing three times, immunoprecipitates were separated by SDS–PAGE, and the gels were used for MS analysis.

LC-MS analysis was performed on a Q Exactive mass spectrometer (Thermo Scientific) coupled to an Easy nLC (Proxeon Biosystems) for 90 min. The peptides were loaded onto a reversed-phase trap column (Thermo Scientific) connected to a C18-reversed-phase analytical column in buffer A (0.1% formic acid) and separated with a linear gradient of buffer B (84% acetonitrile and 0.1% formic acid) at a flow rate of 300 nl/min controlled by IntelliFlow technology. The mass spectrometer was operated in positive ion mode. MS data were acquired using a data-dependent top10 method dynamically choosing the most abundant precursor ions from the survey scan (300–1800 $m/z$) for HCD fragmentation. The dynamic exclusion duration was 40 s, survey scans were acquired at a resolution of 70,000 at $m/z$ 200, the resolution for HCD spectra was set to 17,500 at $m/z$ 200, and the isolation width was 2 $m/z$. The normalized collision energy was 30 eV, and the underfill ratio, which specifies the minimum percentage of the target value likely to be reached at maximum fill time, was defined as 0.1%. The instrument was run with peptide recognition mode enabled. The MS raw data for each sample were searched using the MASCOT engine (Matrix Science) embedded into Proteome Discoverer 1.4 software for identification and quantitation analysis. Significance was assessed with two-sided Student's $t$-tests. The differentially expressed peptides were subsequently filtered for a median fold-change >1.3 and $p$ value < 0.05.

## Animal surgery

All rodent experimental procedures were approved by the Animal Welfare Ethics Committee of Shanghai Jiao Tong University School of Medicine Affiliated Sixth People's Hospital (no. DWLL2023-0522). Rodents used in this study were anesthetized with intraperitoneal administration of pentobarbital. Sample size in each experiment was illustrated in corresponding figure legends.

Adult male C57BL/6 mice (aged 10 weeks) or Sprague-Dawley (SD) rats (aged 12 weeks) sciatic nerve crush injury was modeled by compressing the target nerve area with a total length of 2 mm three times, 20 s each, with #5 forceps. The crush location was marked with India ink, and the CMAP was measured before and after induction of the lesion to ensure efficient axonotmesis. Subsequently, the muscles and skin were sutured, and the animals were allowed to recover in a warm cage. Amoxicillin was administered in the drinking water for one week after surgery.

Adult male C57BL/6 mice (aged 10 weeks) or SD rats (aged 12 weeks) were used for sciatic nerve defect models. After the outside of the right thigh was shaved, the skin was cut, the muscles outside the thigh were separated, and the sciatic nerve was exposed. A segment of the sciatic nerve was removed, leaving a 5-mm nerve gap for mice and a 15-mm gap for rats after contraction of the nerve endings. Then, the nerve conduit was sutured to the proximal and distal nerve stumps using 6-0 sutures. In the autograft group, the nerve defect was bridged by a resected nerve segment with reversed polarity. The muscles were sutured with 4-0 sutures, and the skin was sutured with 3-0 sutures to close the surgical incision. All rodents were housed and fed routinely and euthanized at different time points.

SD rats aged 12 weeks were also used to establish a sciatic nerve transection model. The sciatic nerves were exposed under general anesthesia in aseptic conditions and transected at mid-thigh. Then, the nerves were placed in situ (for MMRN1 guidance assay) or sutured (for MMRN1 blocking assay) using 6-0 sutures. The muscles were sutured with 4-0 sutures, and the skin was sutured with 3-0 sutures to close the surgical incision.

All canine experimental procedures were approved by the Animal Welfare Ethics Committee of Shanghai Jiao Tong University (no. 20230502). Beagles used in this study were anesthetized with intravenous administration of Zoletil-50. Generally, the beagle complicated sciatic nerve defect model was illustrated in Fig. 6a, and a total number of 27 male beagles aged 12 months were divided into 5 groups as SpinMed ($n = 6$), single counterpart ($n = 6$), hollow counterpart ($n = 3$), human-derived decellularized graft ($n = 6$) and autograft ($n = 6$). The individualized SpinMed grafts used in beagle models were customized based on MRI data, and the surgical procedures were illustrated in Fig. 6a, b. Electrophysiological detection and histological examinations were shown below or referred to as rat experiments.

## Walking track assessment

To characterize functional recovery at 8 or 12 weeks after surgery, rats were tested in a confined walkway (10 cm wide by 150 cm long) as previously described[47], and individual footprints highlighted with green fluorescence were recorded ($n = 5$). The following parameters including print length (PL; distance from the heel to the third toe), toe spread (TS; distance from the first to the fifth toes) and intermediary toe spread (IT; distance between the middle of the second and fourth toes) were evaluated. Data were collected for both the experimental (E) and the normal (N) hind legs. The following formula was used to calculate the sciatic function index (SFI) scores: SFI = $-38.3 \times$ (EPL−NPL)/NPL + $109.5 \times$ (ETS−NTS)/NTS + $13.3 \times$ (EIT−NIT)/NIT−8.8.

The SFI scores range from $-100$ to 0, in which $-100$ indicates total impairment, and 0 represents normal function.

## Sensory function tests

We employed hot plate and Von Frey tests to evaluate the thermal and mechanical perception restoration of rats at postoperative 8 or 12 weeks, respectively, as described by previous study[51]. Rats were habituated on a temperature plate at room temperature for 20 min on two consecutive days, and on the testing day, the temperature plate was set at 54 °C (shorter than 30 s). Response to the plate of each rat was recorded, and the latency of paw withdrawal was analyzed. As for the Von Frey test, the plantar surface of rats was stimulated with calibrated von Frey monofilaments (range from 0.008 to 300 g), and the responsive threshold of paw withdraw was determined by Dixon's up-down method followed by stimulus intensity calculation.

## Electrophysiology assessment

To characterize the electrophysiologic properties of rats at post-operative 12 weeks and beagles at postoperative 8 or 12 months, the sciatic nerves of the experimental side were re-exposed under anesthesia. Bipolar hooked stimulating electrodes were applied onto the sciatic nerve trunk at the proximal end, and the time to deflection (latency) and CMAP were recorded on the gastrocnemius belly at the ipsilateral side using an 8-channel physiologic signal recorder (RM-6280C). Both the NCV and CMAP of repaired nerves were recorded and calculated. The experiment was repeated three times in each group.

## FG retrograde tracing

FG retrograde tracing assay was performed to examine the axonal retrograde transport. In brief, 10 µl of 4% FG solution (Fluorochrome Inc.) was injected into the sciatic nerve, 5 mm away from the distal end of the conduits. The rats were allowed a survival of another 5 days and then anesthetized with an overdose of pentobarbital. The spinal cord segments at L5, L6, and S1, together with the L5 to S1 DRGs, were excised and cut into 20-µm-thick transverse sections for spinal cords and longitudinal sections for DRGs, followed by observation under a confocal laser scanning microscope with ultraviolet illumination. The number of FG-labeled motor neurons in the anterior horn of spinal cord sections was counted, while the percentage of FG-labeled sensory neurons in DRG sections was calculated by dividing the FG-positive neuron number by the total neuron number.

## TEM detection

The regenerative nerves were fixed in a solution of 2.5% glutaraldehyde (Solarbio) for 48 h, and then transverse ultrathin sections at a thickness of 50 nm were cut on a microtome (Leica, EM UC6), placed on 0.5% formvar coated meshes and stained with uranyl acetate and lead citrate. Images were captured by a TEM (HITACHI, HT7700 Exalens) with an accelerating voltage of 100 kV. The evaluation of the g-ratio (defined as the inner neurite diameter divided by the outer diameter of the myelinated sheath) was carried out by random fields of each TEM image.

## Histological analyses

The regenerative nerve tissue and muscle of rats and beagles were submitted to H&E staining, TB staining and immunofluorescence staining followed by quantitative analyses. Briefly, the nerve samples were fixed in 4% PFA overnight and subjected to a dehydration process. These samples were then cut into 10 µm-thick cross-sections on a cryostat. The sections were stained with H&E or 1% TB followed by capture under an optical microscope. The primary antibodies against NF200 and Tuj1 were employed to identify regenerated axons at the midpoint of nerves within the conduits. SCs and myelin sheath were identified by S100β and MBP respectively. Secondary antibodies were then applied to incubate the sections. Finally, the nuclei were counterstained with DAPI. The mean densities of myelinated nerve fibers were calculated for each group.

## Statistical analyses

When two groups were compared, a two-tailed Student's $t$-test was used to assess statistical significance. One-way analysis of variance (ANOVA) followed by Tukey's post-hoc tests was used to assess statistical significance when analyzing multiple groups unless otherwise stated. Statistical calculations were performed using GraphPad Prism 9.0, and a $p$-value < 0.05 was considered statistically significant. Detailed statistics were illustrated in corresponding figure legends.

## Reporting summary

Further information on research design is available in the Nature Portfolio Reporting Summary linked to this article.

# Data availability

The main data supporting the results of this study are available within the paper and supplementary information. The proteomics and mass spectrometry data are available from the Mendeley Data (https://data.mendeley.com/datasets/hmd4fpnv9j/2). Any additional requests for information can be directed to and will be fulfilled by the corresponding authors. Source data are also provided with this paper. Source data are provided with this paper.

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

## Acknowledgements

This work was supported by the National Key R&D Program of China (No. 2021YFC2400800 by C.F.), National Natural Science Foundation of China (No. 82330076 by C.F., 82372409 by Y.Q., 52173117 and 21991123 by Z.Y.), State Key Laboratory for Modification of Chemical Fibers and Polymer Materials (No. KF2308 by Y.Q.) and Excellent Youth Cultivation Program of Shanghai Sixth People's Hospital (No. ynyq202201 by Y.Q.). We acknowledge BioRender.com for providing icons of illustrations, and Springer Nature Author Services for language editing.

## Author contributions

L.K., Y.Q., Z.Y., and C.F. conceived and designed the study. L.K., X.G., and X.Y. performed most of the experiments and analyzed the data. H.X.

assisted with molecular dynamics simulation. Q.K., W.S., Y.Q., and Z.Y. provided suggestions about the project design and data presentation. L.K., W.S., and Y.Q. wrote the original draft manuscript, and Z.Y. and C.F. revised the manuscript with input from all authors.

## Competing interests

The involved techniques as an invention patent in China were submitted to State Intellectual Property Office (No. 202310198696.2), and an international patent (PCT) to World Intellectual Property Organization Office (No. PCT/CN2023/083644). The authors declare no other competing interests.
