## [Peer Review File · Nature Communications]

REVIEWER COMMENTS

Reviewer #1 (Remarks to the Author):

The authors designed a sugar painting-inspired individualized multilevel neurium-mimetic nerve graft named "SpinMed" composed of modified epineurium, topological perineurium, and spontaneously formed endoneurium (SpinMed) and evaluated the regenerative potential in various nerve defect models including rats and beagles. The results showed the rapid and individualized fabrication of SpinMed graft based on anatomy and digital imaging, possessed enhanced mechanical properties, successfully accelerated vascular extension, and improved nerve regeneration with multiple endpoints in rat and beagle nerve defect models. The design is novel, and the SpinMed" demonstrated promising therapeutic outcomes in peripheral nerve regeneration and biosafety for the ongoing nonhuman primate model and clinical trial. However, several issues should be addressed regarding the manuscript.

1. In Extended Data Figure 1, b and C showed n = 23, however only three patients' samples in a. It needs to be clear whether the total patient number is 23 or the total slide sample is 23.
2. Fig 2e defect model length should be indicated, and quantitative analyses are needed.
3. In Figure 2h and 4f WB, Actin bands should be provided.
4. In Figure 4f, not sure if there is a splicing process for the WB images, practically for the 2nd and 6th rows.
5. In Figure 5f, the expanded parts in the "zoom in" panel should be marked in the left column.
6. In Figure 5i, the spinal cords were not sectioned at the same level. Nestin and GFAP were labeled without quantification. Fluoro-Gold retrograde tracing was included in the method section but related results were unclear in the results section. These results should be clearly presented and will further enhance the quality of the manuscript.
7. The canine experimental procedures are unclear to show whether it's a bifurcation injury model (such as in Figure 6). The procedure details such as the distal ends of the defect section should be provided.
8. In Figure 6, What does "sing" mean, a straight empty scaffold? It's unclear whether autografts are linear or bifurcation. Representative figures for four groups need to be provided.
9. The representative figure in Fig 6a shows that the decellularized scaffold is a linear graft. It's unclear whether the defect injury is the same for all groups, as it's unclear how to suture a linear straight scaffold ("sing", decellularized, or autograft) to two distal ends from the bifurcation defect injury created for the individualized bifurcation SpinMed graft.

10. It's not appropriate to compare a linear scaffold to the individualized SpinMed graft. An empty bifurcation scaffold should be used as a control instead.
11. Fig 6G, quantification data needs to be provided.
12. Supplementary Fig. S7 b, the quality of staining needs to be improved (Dead staining in the 2nd and 3rd row is unclear).
13. A 15-mm-long defect model was created for "male C57BL/6 mice (aged 10–12 weeks) or Sprague-Dawley rats: in the method section, however, a 15-mm-long defect is too long for the mice.
14. In the animal experiment section, the total detailed numbers of species used in this study should be clear. The number should also be clear in all figures and supplemental figures (some of them have had the numbers).
15. The authors claimed that "integrating biomimetic layouts into grafts based on neural structures and the regenerative microenvironment and preventing nerve fiber mismatching" and "enables regenerative nerves to extend with less mismatching" in the discussion. However, such comparisons are not supported by the study design and results. It's unclear how this design will promote motor-to-motor or sensory-to-sensory matching in the neurium-mimetic fabrication of grafts. Thus, should be tuned down.

Minor:

1. Abbrs should have the full name first. For example, MMRN1 and PECAM1 et al. In Fig 1 J: Abbrs should be consistent with other panels 9 or with a full name first in the figure legend, suggest using the same full name. In addition, the full name for TCP in Figure 2 and other figures was missing.
2. In Fig 6, the four groups should have full names in the legend (or in the figure).
3. The rationale for using the HEK293T cell line needs to be briefly provided.
4. The scale bar is 10um for Figure 6G (beagles) and 2um for Figure 5F (rats), which seems appropriate. Please double-check them for accuracy.
5. "SpinMed is likely to be regulated as a class III device (510(k) in the USA and NMPA in China) in the future" should be acknowledged in other places per journal style instead (but not in the conclusion part).

Reviewer #2 (Remarks to the Author):

The manuscript describes a novel nerve graft/device using biomaterial technology advances this reviewer is less familiar with, but seem to be highly novel and an advance for the field. These devices/technologies are then assessed using in vitro and cell culture assays, and animal models. There are concerns regarding the suitability of the cell culture assays used, where the methods are not clear if they recapitulate the goal the in vivo device is trying to solve. Furthermore, the in vivo experimental design does not address or demonstrate the major accomplishments / outcomes shown from the in vitro experiments to understand if these were achieved in vivo, such as vascular ingrowth and alignment, along with Schwann cell and axon coupling. Overall, there are major concerns that need to be addressed in the manuscript. For the field as a whole, the biomaterial technology advanced is novel and interesting for the field, but the achievement in regeneration / outcomes is not clear because the control employed lack rigor. Detailed criticism is as follows:

1. The in vitro / cell culture assays have methodology that lacks sufficient detail. There is concern that these assays do not mimic the in vivo implanted devices sufficiently, as a major hurdle for the field is achieving nerve mimicry at large / longitudinal scales. The methodology makes it unclear whether these culture assays provided enough of a simulation to the in vivo scenario to interpret these results an advance for this device.
2. There is no data demonstrating degradation of the product or by-products in vivo. Is the device cleared by the time of evaluation? What are the long-term implications?
3. The use of mice and rats is confusing as written in the methods. The methods as written cannot be correct as 15 mm defects in mice sciatic nerve is not possible. It wasn't clear when mice were used in the results.
4. In rats, the walking track behavioral assessment is useful and should be included, but it has many major shortcomings. It should not be the only behavioral (functional) assessment used. Specific sensory testing should be evaluated (mechanical, allodynia or pain) and additional sensorimotor testing is fairly standard for new data generated in this field.
5. A major flaw is the use of a single endpoint in rat and canine models. To keep animal numbers low, this is understandable, but behavioral (functional) evaluation should be assessed over time, not at a single endpoint.
6. Additionally to point #5 and as stated above in the major summary, a single endpoint well past early regeneration limits the understanding of how these devices affected early regeneration, such as vascular outgrowth and alignment, Schwann cell function and properties, and initial axon outgrowth and coupling.
7. The retrograde labeling results for rodents were confusing. Usually this method allows quantification of neurons (sensory vs motor) regenerating axons, as parent regenerating axons can branch. Therefore, axon counts as an outcome metric alone can lead to results that are misleading, as axons that branch more can lead to an outcome suggesting higher axons counts and thus superior regeneration, when in fact fewer neurons might have regenerated axons. Therefore, the use of this metric by the authors is ideal. However, the methods describe counting neurons, but the results do not report this. Instead, the results describe the use of nanoparticle quantity as a metric, which is misleading and not helpful, as more axon branching could lead to increased accumulation of nanoparticles. It is also unclear what

increased nanoparticles means as the methods do not describe how this could be evaluated. The extended data shows neuron counts, but these are much too low of values to reflect neuron quantity regenerating axons in rats (rats have ~20,000 sensory neurons and ~2,000 motor neurons). The methods and results needs attention to understand these data as presented.

8. The experimental design transitioning from the dog vs rat model is confusing and not rationale. Why were different controls used in the rat compared to the dog? The use of branching alone is not a sufficient reason.

9. The acellular nerve allograft used for the canine is not described and critical to understand. Was it generated from canine or was a human product (AxoGen) used? The methods used to generate this control could yield a product that would not facilitate robust regeneration and could induce an immune response, making it an inaccurate control.

10. The dog paradigm of repair is interesting, but not clinically relevant. A sciatic nerve is almost always a cable repair to reflect different branches. The autograft used is not the standard of care – cabled autograft repair is.

11. The dog studies do not provide sufficient methodological detail. The experiments are not the same as rodent, where the functional (behavioral) assay is completely different. Without more detail, this cannot be assessed. As well, it is unclear why the histology in this model was only partially quantified. Axon quantity should be reported.

12. The statistics is unclear. ANOVA is used for multiple groups, but no post-hoc methods are provided.

Reviewer #1 (Remarks to the Author):

The authors designed a sugar painting-inspired individualized multilevel neurium-mimetic nerve graft named “SpinMed” composed of modified epineurium, topological perineurium, and spontaneously formed endoneurium (SpinMed) and evaluated the regenerative potential in various nerve defect models including rats and beagles. The results showed the rapid and individualized fabrication of SpinMed graft based on anatomy and digital imaging, possessed enhanced mechanical properties, successfully accelerated vascular extension, and improved nerve regeneration with multiple endpoints in rat and beagle nerve defect models. The design is novel, and the SpinMed” demonstrated promising therapeutic outcomes in peripheral nerve regeneration and biosafety for the ongoing nonhuman primate model and clinical trial. However, several issues should be addressed regarding the manuscript.

1. In Extended Data Figure 1, b and C showed $n = 23$, however only three patients’ samples in a. It needs to be clear whether the total patient number is 23 or the total slide sample is 23.

Response: Thanks for your kind suggestion. We are sorry for the unclear description of human specimens in the initial version. **We confirmed that the nerve samples were collected from 23 patients including 16 males and 7 females**, for which we reorganized the relevant expression in the “Methods” and “Results” sections as “...we acquired 23 nerve tissue specimens from 23 independent patients (16 male and 7 female individuals; age ranging from 31 to 56 years) who had suffered amputation due to posttraumatic complication” (Line 498 to 500, Page 24), and “...in clinical practice, we found that areas of the posttraumatic PNR were often destroyed following cicatricial fibrosis as identified in 23 patients due to the weak epineurium, making it difficult to keep out adjacent tissue” (Line 102 to 104, Page 4). Detailed information about the 23 specimens (each specimen from an independent patient) has been shown in the below list, which was also deposited into “Supplemental Materials” (revised Supplementary Table 1).

Supplementary Table 1. Human specimen information.

Patient	Gender	Age	Specimen type
1	male	37	tibial nerve
2	male	42	sciatic nerve
3	male	31	tibial nerve
4	male	38	ulnar nerve
5	male	34	tibial nerve
6	female	54	tibial nerve
7	male	36	common peroneal nerve
8	male	39	tibial nerve
9	male	47	ulnar nerve
10	female	49	sciatic nerve
11	male	45	tibial nerve branches
12	female	51	tibial nerve
13	male	46	tibial nerve
14	male	52	ulnar nerve
15	male	35	common peroneal nerve
16	female	54	tibial nerve
17	female	47	tibial nerve
18	male	34	tibial nerve
19	male	51	tibial nerve
20	male	56	tibial nerve
21	male	37	common peroneal nerve
22	female	41	tibial nerve
23	female	38	tibial nerve

2. Fig 2e defect model length should be indicated, and quantitative analyses are needed.

Response: Thanks for your kind suggestion, and we are sorry to miss a clear

label in Fig. 2e. **The mice (5 mm) and rat (15 mm, image not completely shown) nerve defects have been marked by lines and arrows in the revised figure** followed by quantification (as shown below), as well as modifications for figure legends. The results showed that the biomimetic architecture of SpinMed accelerated the progress of vascularization from the proximal stump to the distal terminal in both mice and rats.

Legends (revised Fig. 2e, f): Representative images of sciatic nerves harvested from mice or rats 2 weeks after implantation (white full lines represent initial stump terminal, and arrows indicate extended vessels) (left) followed by quantification of vascular extension rates (right) (n = 6). Mean values are shown and error bars represent \pm s.d. as analyzed by t-test.

3. In Figure 2h and 4f WB, Actin bands should be provided.

Response: Thanks for your comments. In previous Fig. 2h, we employed the Coomassie Blue staining to confirm the equal protein volume among various groups. The co-IP assay in Fig. 4f aimed to verify the bindings between MMRN1 and HMMR, MMRN1 and CDK1, or HMMR and CDK1, rather than their relative expression levels. Therefore, we have added internal reference protein bands (GAPDH) of these protein samples followed by updated quantification in revised Fig. 2i, j (as shown

below) to eliminate misunderstandings, and we have reorganized the “Results” descriptions of co-IP assay in Fig. 4f (Line 267 to 270, Page 12).

Legends (revised Fig. 2i, j): Western blot images of MMRN1 expression in vascular endothelial cells (left) and quantification of the MMRN1/GAPDH ratio (right) (n = 3). Mean values are shown and error bars represent \pm s.d. as analyzed by t-test.

4. In Figure 4f, not sure if there is a splicing process for the WB images, practically for the 2nd and 6th rows.

Response: Thanks for this kind comment. **We declare that there was no any cutting or splicing blot in this study.** The illuminating artifacts (seem like splicing signs) presented on the second and sixth rows in the Fig. 4f panel may be caused by rapid electrophoresis rate and uncompleted blocking, and we now provide the raw images below without any modifications to address this concern. Moreover, we repeated this co-IP assay, and the results strongly supported the current findings.

Legends (raw images for revised Fig. 4f): left image relevant to 2nd blot, and right one relevant to 6th blot.

5. In Figure 5f, the expanded parts in the “zoom in” panel should be marked in the left column.

Response: Thanks for your kind suggestion. We have marked the expanded parts of low-magnification images in the left column by using red rectangular boxes (as shown below), which helps to understand this part.

Legends (revised Fig. 5f): Representative TEM images of regenerative nerve tissue (transverse view, left panel) and zoom in images (right panel) from various groups (n = 8).

6. In Figure 5I, the spinal cords were not sectioned at the same level. Nestin and GFAP were labeled without quantification. Fluoro-Gold retrograde tracing was included in the method section but related results were unclear in the results section. These results should be clearly presented and will further enhance the quality of the manuscript.

Response: Thanks for this constructive comment. All spinal cord specimens were sectioned at near L5 levels, but it seemed that they were not from the same levels due to sample collection deviation and individual differences in cutting procedures. **We have re-performed section and staining procedures by using the**

spinal cord specimens followed by labels for gray matter morphology and updated quantification (as shown below), so as to meet the requirement of consistent section level as far as possible.

Legends (revised Fig. 5i; revised Supplementary Fig. S9): Representative immunofluorescence images of Nestin and GFAP expression in the spinal cord (L5) (top panel) followed by quantification (n = 5) (bottom panel). Mean values are shown and error bars represent \pm s.d., as analyzed by one-way ANOVA with Tukey's post hoc tests.

The Fluoro-Gold retrograde tracing assay was repeated at postoperative 12 weeks, in which the revised results and images were updated in the revised figures and manuscript, and relevant descriptions of retrograde tracing assay methodology and findings have been supplied in the "Methods" and "Results" sections (Line 333 to 341, Page 16; Line 772 to 781, Page 32).

Legends (revised Extended Data Fig. 6h, i): Representative images of FG-labeled neurons observed in DRG sensory neurons or motoneurons within AH (left panel), and quantification for the number of FG-labeled neurons (n = 5) (right panel). Mean values are shown and error bars represent \pm s.d., as analyzed by one-way ANOVA with Tukey's post hoc tests.

7. The canine experimental procedures are unclear to show whether it's a bifurcation injury model (such as in Figure 6). The procedure details such as the distal ends of the defect section should be provided.

Response: Thanks for this constructive comment. We are sorry for the unclear expression and illustration in the initial version. **Now we have provided the scheme and surgical images for each group as shown below and revised manuscript**, and we believe the illustration and modification could contribute to a better understanding of canine surgical methods and procedures. The revised images were also presented in Fig. 6a, b and "Extended Data" of the updated manuscript.

Legends (revised Fig. 6a, b): Illustration of surgical procedures performed in complicated sciatic nerve defects in the “Single” “SpinMed” “Decellularized graft” and “Autograft” groups (a), and surgical images of application methods in various groups, with labels for sciatic nerve (1), tibial nerve (2) and common peroneal nerve (3).

8. In Figure 6, What does “sing” mean, a straight empty scaffold? It’s unclear whether autografts are linear or bifurcation. Representative figures for four groups need to be provided.

Response: Thanks for your kind suggestion. First, the “Sing.” in the initial manuscript referred to a straight SpinMed counterpart with multilevel neurium-mimic structures, instead of a hollow device. Now we have provided the scheme and surgical images for each group in the revised version. As for autograft, we employed the cable suture strategy using the *in situ* nerve tissue after reversing, by which the epineurium of the distal terminal (branching site, tibial nerve and common fibular nerve) was partially cut open and side-to-side sutured into a robust nerve trunk, while the proximal sciatic nerve was split along the longitudinal axis to two cables with appropriate ratios on demand (*Neural Regen Res*, 2017, PMID: 29323049; *JAMA Facial Plast Surg*, 2016, PMID: 27197116). Then, we performed end-to-end anastomosis to achieve proximal and distal stump connection for mimicking cable sutures in clinical practice to the maximum extent. **Illustrations and key surgical procedures about several groups of animal models are shown above (#7).**

9. The representative figure in Fig 6a shows that the decellularized scaffold is a linear graft. It's unclear whether the defect injury is the same for all groups, as it's unclear how to suture a linear straight scaffold ("sing", decellularized, or autograft) to two distal ends from the bifurcation defect injury created for the individualized bifurcation SpinMed graft.

Response: Thanks for your kind comment. We are sorry for the unclear descriptions of surgical procedures of the canine nerve defect model. **The defect injury was the same for all groups (bifurcation defect, revised Fig. 6a, b), but repair methods were distinct among several groups.** As illustrated in schemes and surgical images (images shown above, #7), the decellularized scaffold with various sizes and autografts was integrated into an appropriate graft by care-cable sutures. We believe the revised scheme is easily understood by reviewers and readers, in which **nerve cable sutures as the primary methods of autograft and decellularized graft implantation are displayed.** As confirmed by previous studies, nerve polarity tended nearly not to influence the nerve graft outcomes (*J Reconstr Microsurg*, 2002, PMID: 12177822; *Microsurgery*, 2017, PMID: 27935644), and therefore, we inverted the defected nerve (180°) to mimic autograft from other body sites, such as sural nerve, for nerve integrity restoration (*Nat Commun*, 2018, PMID: 29358641; *Biomaterials*, 2020, PMID: 32554132). **This method is clinically relevant and the most suitable control model for the autograft and/or commercial decellularized graft in this study.**

10. It's not appropriate to compare a linear scaffold to the individualized SpinMed graft. An empty bifurcation scaffold should be used as a control instead.

Response: Thanks for your constructive comment. In this study, an individualized hollow bifurcation graft was initially designed as a SpinMed counterpart to repair an identical nerve defect model (n = 3), but **it failed to repair critical sciatic nerve defects longer than 4cm owing to slow neuronal growth**, also found in rat huge nerve defect models. Segmental nerve defect and slow restoration, along with serious Wallerian degeneration at an early stage, altogether led to nerve

regeneration failure observed from less than 4-month follow-up (*EMBO J*, 2019, PMID: 31268609; *Neural Regen Res*, 2014, PMID: 25206870). During this period, **denervation has always caused ulcers and even incurable tissue defects on the suffered limbs of beagles in this group (shown below), and the suffered beagles failed to walk and even had difficulty standing, for which ethics committee recommended us to cease this group test and give them euthanasia.** Therefore, we did not present relevant data and long-term follow-up of this blank control group regarding the above reasons. Despite this, here we provide the gross view images of beagles suffering repair failure in the SpinMed hollow counterpart group.

Legends (Review Report Fig. 1): Representative images of tissue defect feet (left) of beagles receiving hollow counterpart implantation, and nearly intact feet (right) of beagles in SpinMed group.

Importantly, **we also presented TEM images of the regenerative nerve within P1 and P2 at postoperative 16 weeks**, in which we failed to acquire the specimens within P3 and P4 due to unavailable tissues (empty or minimum). These findings have been also added to the revised Extended Data Fig. 7d, e for clearly illustrating differential outcomes in the hollow counterpart group as a blank control.

Legends (revised Extended Data Fig. 7d, e): Illustration of the local nerve model and positions of TEM detection (top panel), where transverse ultrastructural views were obtained (bottom panel) (n = 3).

11. Fig 6G, quantification data needs to be provided.

Response: We are sorry for the unclear expression in the “Results” section. This part has been displayed in the revised Extended Data Fig. 7e, f, followed by **quantification for the *g*-ratio and myelin sheath thickness as shown below.**

Legends (revised Extended Data Fig. 7e): Quantitative analysis for myelin sheath thickness and g-ratio within P2 to P4 at postoperative 16weeks (n = 3). Mean values are shown and error bars represent \pm s.d., as analyzed by one-way ANOVA with Tukey's post hoc tests.

12. Supplementary Fig. S7b, the quality of staining needs to be improved (Dead staining in the 2nd and 3rd row is unclear).

Response: Thanks for your kind comment. We are sorry for the unlabeled images in Supplementary Fig. S7b. Dead cells were rare in several groups of the present study. However, **we have repeated this experiment followed by quantification as shown below**, and the results were updated in the new manuscript with labels by white arrows (2nd row). We believe the new one is better to present the results of live/dead cell staining.

Legends (revised Supplementary Fig. 7b, c): Live or dead cells stained by calcein-AM or propidium iodide (PI) at 72 h (left panel) and quantification of dead/live cell ratio (n = 4) (right panel). Mean values are shown and error bars represent \pm s.d., as analyzed by one-way ANOVA.

13. A 15-mm-long defect model was created for “male C57BL/6 mice (aged 10–12 weeks) or Sprague-Dawley rats: in the method section, however, a 15-mm-long defect is too long for the mice.

Response: Thanks for your kind comment. We are sorry for the ambiguous descriptions of rodent models, especially the unclear distinction between rats and mice. Now we have revised it in the “Methods” and “Results” sections for accurate expression (Line 174 to 178, Page 7; Line 719 to 721, Page 30 to 31). **The mice sciatic nerve defect model (5-mm defect in this study) was only used in the revised Fig. 2e to investigate the effects of topological cues on axially vascular extension.** In the revised figures, the defect zones of both mice and rats have also been marked by white lines followed by quantification as shown below.

Legends (revised Fig. 2e, f): Representative images of sciatic nerves harvested from mice or rats 2 weeks after implantation (white full lines represent initial stump terminal, and arrows

indicate extended vessels) (left) followed by quantification of vascular extension rates (right) (n = 6). Mean values are shown and error bars represent \pm s.d. as analyzed by t-test.

14. In the animal experiment section, the total detailed numbers of species used in this study should be clear. The number should also be clear in all figures and supplemental figures (some of them have had the numbers).

Response: Thanks for your kind suggestion. We are sorry that parts of detailed sample sizes were missed in the initial version, but **we have now added specific numbers of species in each experiment into figure legends**, such as n = 5 (five independent animals in each group), and the “Methods” section of the revised manuscript (Line 710, Page 30).

15. The authors claimed that “integrating biomimetic layouts into grafts based on neural structures and the regenerative microenvironment and preventing nerve fiber mismatching” and “enables regenerative nerves to extend with less mismatching” in the discussion. However, such comparisons are not supported by the study design and results. It’s unclear how this design will promote motor-to-motor or sensory-to-sensory matching in the neurium-mimetic fabrication of grafts. Thus, should be tuned down.

Response: Thanks for your constructive comment. **i)** First, “match” in the initial manuscript referred to geometrical guidance rather than sensory and/or motor fiber type-dependent repair, and therefore this term may bring misunderstanding to readers and tend to be replaced in several “nerve fiber”-relevant sentences. **ii)** Generally, the mammal sciatic nerve contains 93% sensory fibers and 7% motor nerve fibers, and in most peripheral nerve regeneration studies, it is difficult to distinguish sensory fiber type and motor nerve fiber, especially at the early stage, within regenerative neural tissue. Instead, they are often evaluated by sensory and/or motor function detection (Nat Biomed Eng, 2018, PMID: 30948823; Nat Mater, 2024, PMID: 37814117). In the present study, we also primarily employed sensory and motor behaviour tests to evaluate their functional restoration. **iii)** We are investigating the regenerative

preferences of both sensory and motor fiber types, and based on it we are seeking for specific tissue engineering strategies in the future, also known as motor-to-motor or sensory-to-sensory matching. Taken together, this study did not engage in specific sensory and/or motor nerve fiber matching, and we have modified relevant descriptions and discussions in the revised manuscript as follows.

“... integrating biomimetic layouts into grafts based on neural structures and the regenerative microenvironment to induce injured nerve fiber extending ...” (Line 54 to 56, Page 2), “Precise nerve fiber guidance by biomimetic graft is undoubtedly helpful for promoting efficient neural regrowth ” (Line 57 to 58, Page 2), “... Aligned filler with a less than 100- μ m cue was designed to simulate the perineurium as an efficient adhesive media, guiding oriented extension of the nerve bundle in a higher level of regenerative efficiency ...” (Line 427 to 429, Page 21), and “... enables regenerative nerves to extend with axial guidance ...” (Line 461 to 462, Page 22).

Minor:

1. Abbrs should have the full name first. For example, MMRN1 and PECAM1 et al. In Fig 1 J: Abbrs should be consistent with other panels 9 or with a full name first in the figure legend, suggest using the same full name. In addition, the full name for TCP in Figure 2 and other figures was missing.

Response: Thanks for your kind comment. There were several missing full names when abbreviations first occurred, but **we now have supplemented them and screened all abbreviations for their full explanations and normative formats.** For example, “natural nerve (NN)” “decellular nerve (DN)” “multimerin 1 (MMRN1)” “platelet endothelial cell adhesion molecule 1 (PECAM1)” “tissue culture plate (TCP)” “polycaprolactone (PCL)” “dorsal root ganglion (DRG)”.

2. In Fig 6, the four groups should have full names in the legend (or in the figure).

Response: Thanks for your kind comment. **We have modified the group names and illustrated their full names in figure legends** as “Holl.” (hollow counterpart), “Sing.” (single counterpart), “Spin.” (SpinMed), “Decell.” (decellularized graft) and

“Auto.” (autograft).

3. The rationale for using the HEK293T cell line needs to be briefly provided.

Response: We are sorry for the unclear descriptions in the “Methods” and “Results” sections. Human embryonic kidney (HEK) 293T cell is known as a relatively simple and efficient transfection system that allows for rapid and scalable production of target protein following plasmid-mediated gene delivery. A substantial number of studies employed the HEK293T cell line to confirm specific RNA sequences and multi-protein interaction relations (*Nature*, 2024, PMID: 38297130; *Cell Metab*, 2022, PMID: 35705079). **We have explained it in the “Results” section in the revised version**, as “We further employed the co-IP assay to confirm the interactions among MMRN1, HMMR and CDK1 in the HEK293T cell line, a transformed human embryonic kidney cell line, verifying the ligand-receptor relationships between MMRN1 and HMMR, MMRN1 and CDK1, as well as HMMR and CDK1” (Line 267 to 270, Page 12).

4. The scale bar is 10um for Figure 6G (beagles) and 2um for Figure 5F (rats), which seems appropriate. Please double-check them for accuracy.

Response: Thanks for your kind comment. We have checked relevant TEM raw data of beagles and rats, and we declare that the scale bars are accurate. However, **we have changed the sizes of scale bars in the two figure panels for consistent presentations (scale bars, 5 μm)**, as well as additional data in the revised manuscript.

5. “SpinMed is likely to be regulated as a class III device (510(k) in the USA and NMPA in China) in the future” should be acknowledged in other places per journal style instead (but not in the conclusion part).

Response: Thanks for your kind suggestion. The relevant descriptions of clinical translation of SpinMed have been removed from the conclusion part (Line 483 to 486, Page 23).

Reviewer #2 (Remarks to the Author):

The manuscript describes a novel nerve graft/device using biomaterial technology advances this reviewer is less familiar with, but seem to be highly novel and an advance for the field. These devices/technologies are then assessed using in vitro and cell culture assays, and animal models. There are concerns regarding the suitability of the cell culture assays used, where the methods are not clear if they recapitulate the goal the in vivo device is trying to solve. Furthermore, the in vivo experimental design does not address or demonstrate the major accomplishments/outcomes shown from the in vitro experiments to understand if these were achieved in vivo, such as vascular ingrowth and alignment, along with Schwann cell and axon coupling. Overall, there are major concerns that need to be addressed in the manuscript. For the field as a whole, the biomaterial technology advanced is novel and interesting for the field, but the achievement in regeneration/outcomes is not clear because the control employed lack rigor. Detailed criticism is as follows:

1. The in vitro/cell culture assays have methodology that lacks sufficient detail. There is concern that these assays do not mimic the in vivo implanted devices sufficiently, as a major hurdle for the field is achieving nerve mimicry at large/longitudinal scales. The methodology makes it unclear whether these culture assays provided enough of a simulation to the in vivo scenario to interpret these results an advance for this device.

Response: Thanks for your constructive comments. We are sorry for the insufficient descriptions of the details of in vitro assays, which potentially bring some ambiguous understandings for reviewers and readers. **Generally, these in vitro experiments were employed to evaluate cell behaviours (such as cell proliferation and migration) and cell communication functions (such as paracrine function and intercellular action).** As you said, a major limitation in the nerve regeneration field is that difficult to mimic in vivo morphology and regenerative microenvironment at large scales. Most SpinMed outcomes were acquired from rat and beagle experiments in this study, while in vitro assays were primarily performed for identifying cell activities upon SpinMed surfaces. **We tried to investigate angiogenesis and vessel-myelin coupling within SpinMed at multiple microscales,**

all of which contributed to a global landscape of neural repair in the present study (*Science*, 2022, PMID: 35549409; *Nat Metab*, 2020, PMID: 32778834; *Nat Commun*, 2016, PMID: 27435623). Here, we provide more exhaustive experimental purposes and designs of in vitro assays as following descriptions.

Myofibroblasts were employed to execute a transwell assay across cores of 8.0 μm for detecting the permeability of the SpinMed sheath (revised Extended Data Fig. 2g- i). It aimed to show the epineurium-like sheath of SpinMed exhibiting high permeability to nanoscale proteins and lower permeability to myofibroblasts. This in vitro experiment demonstrated the characterizations of the enhanced perineurium-like structure of SpinMed.

The in vitro proliferation and migration assays of HUVECs were designed to simulate SpinMed filler that provided axially topological cues for vascular growth despite most tests performed on two-dimensional interfaces, in which the proliferative and migrated capacities were examined when HUVECs cultured on SpinMed and its counterparts (revised Fig. 2a, b; Extended Data Fig. 3a, b), as well as tube formation test mimicking in vivo vascular sprout (revised Fig. 2c, d).

To investigate the vessel-nerve coupling mechanisms, high-throughput techniques including proteomics were employed to screen HUVECs-derive paracrine molecules, and then we found the pivot protein MMRN1 functioning in a series of activities (revised Fig. 2g, h). According to it, Western blot (revised Fig. 2i, j), computational simulations (revised Fig. 2k), and immunostaining (revised Fig. 2l) altogether confirmed the crucial roles of MMRN1 in vessel-myelin interaction.

The HUVECs and SCs coculture system was used to directly investigate the intercellular crosstalk that SpinMed anchor-induced vessel-myelin coupling, in which HUVECs were seeded on TCP, random or topological (SpinMed) interfaces and cocultured with SCs to investigate the effects of paracrine factor MMRN1 derived from HUVECs on SCs (revised Fig. 4b, c, f; revised Extended Data Fig. 4b-e). After that, receptors on SCs were verified to receive MMRN1 ligand and induce intracellular signaling transduction (revised Extended Data Fig. 4f-k).

The DRG neurites extension assay was designed to mimic axonal extension

induced by remyelination under in vivo specific interfaces (revised Fig. 4g, h). DRGs were isolated and cultured in neurobasal medium. After 24 h, the medium was then changed to 50% neurobasal medium plus 50% SCs conditioned medium from above “HUVECs-SCs” co-culture systems within various substrates. After 3 days, the average DRG axonal lengths were calculated and analyzed (*Nat Commun*, 2020, PMID: 33257677; *Biomaterials*, 2021, PMID: 33813259). The results reflected axonal extension after vessel-myelin coupling within SpinMed.

Taken together, these in vitro studies were designed to mimic in vivo neural repair microenvironment to a maximum extent, and used to investigate intercellular crosstalk in a more accessible manner. Importantly, the in vitro findings supported animal experiment outcomes. Moreover, we have modified the detailed methodology for improving the scientific rigor of in vitro experimental design, and in the future, this common issue in the neural regeneration field will be well addressed possibly by the development of organs-on-chips.

2. There is no data demonstrating degradation of the product or by-products in vivo. Is the device cleared by the time of evaluation? What are the long-term implications?

Response: Thanks for your kind comments. To address the concerns about the impact of device degradation products, we evaluated it and investigated long-term in vivo status as follows. **i)** First, we have taken photos at postoperative 8 and 12 months in a beagle nerve defect model. The SpinMed seemed to achieve almost complete degradation after 12 months of in vivo implantation. **Only partial sheath residue was observed after 8 months, and disappeared after 12 months as shown below.** In addition, there was no obvious inflammation and immune rejection phenomenon observed within the topical nerves, muscles and stromal tissue. **ii)** In rat and beagle sciatic nerve defect models, we recorded the conduit weight changes (degradation) during the follow-up period as shown below. For the rat model, the SpinMed conduit degraded from 100% weight (day 0) to 77.0% ± 3.7% (4 weeks), 56.2% ± 3.5% (8 weeks) and 38.4% ± 3.9% (12 weeks). For the beagle model, the SpinMed degraded from 100% weight (day 0) to 46.4% ± 3.0% (16 weeks) and **0.7% ± 0.65% (12**

months). These findings showed that SpinMed self-degradation within the implantation zone was confirmed, and the degradation during tissue regeneration did not impact SpinMed integrity. **iii)** An appealing translational feature of SpinMed is that most components, including PCL and silk fibroin, are FDA-approved and favorable for clinical translation, exhibiting excellent biocompatibility and optimized biodegradability. Moreover, the degradation products and/or by-products, such as amino acids and polypeptides, are expected to be water-soluble and furnish newly regenerative tissue (*Biomaterials*, 2021, PMID: 33002786; *Sci Adv*, 2023, PMID: 38117891). **iv)** We further evaluated the biosafety of SpinMed by **detecting histology of global organs and the levels of aspartate aminotransferase (AST), alanine aminotransferase (ALT), lactate dehydrogenase (LDH) and blood urea nitrogen (BUN) in serum at postoperative 12 months**, and no aberrant result was found, which indirectly reveals the biocompatibility of SpinMed degradation product and by-product. Therefore, the above evidence collectively demonstrates the long-term biocompatibility and degradation pattern of SpinMed.

Legends (Review Report Fig. 2): The images of 8-month (left) and 12-month (right) SpinMed implantation.

Legends (revised Supplementary Fig. S11): Degradation ratio of SpinMed after implantation

into rats (total 12 weeks) and beagles (total 12 months) (n = 3). Mean values are shown and error bars represent \pm s.d.

Legends (revised Supplementary Fig. S11): The AST, ALT, LDH and BUN concentrations in serum of beagles detected at postoperative 12 months (n = 3). Mean values are shown and error bars represent \pm s.d., as analyzed by one-way ANOVA.

Legends (revised Supplementary Fig. S11): HE staining for major functional organs at 12 months after SpinMed implantation into beagles (n = 3).

3. The use of mice and rats is confusing as written in the methods. The methods as written cannot be correct as 15 mm defects in mice sciatic nerve is not possible. It wasn't clear when mice were used in the results.

Response: Thanks for your kind comment. We are sorry for the confusing descriptions of rodent models. Now we have revised the unclear presentations involving rats and mice both in the “Methods” and “Results” sections. **The mice sciatic nerve defect model (5-mm defect in this study) was only used in Fig. 2e to**

investigate the effects of topological cues on axially vascular extension, and mice nerve crush models were used in Extended Data Fig. 1 for investigating the associations of α SMA with neural regeneration. In addition, the term “rodent” has been replaced in most paragraphs.

In the revised Fig. 2e, the defect zones of both mice and rats have been marked by white lines, and the corresponding results were also shown in the “Results” section (Line 174 to 178, Page 7).

Legends (revised Fig. 2e, f): Representative images of sciatic nerves harvested from mice or rats 2 weeks after implantation (white full lines represent initial stump terminal, and arrows indicate extended vessels) (left) followed by quantification of vascular extension rates (right) ($n = 6$). Mean values are shown and error bars represent \pm s.d. as analyzed by t-test.

4. In rats, the walking track behavioral assessment is useful and should be included, but it has many major shortcomings. It should not be the only behavioral (functional) assessment used. Specific sensory testing should be evaluated (mechanical, allodynia or pain) and additional sensorimotor testing is fairly standard for new data generated in this field.

Response: Thanks for your kind suggestion. We repeated the rat sciatic nerve

defect experiments according to the 3R (reduction, replacement, and refinement) principle, and **performed non-invasive sensory function examinations at postoperative 8 weeks and 12 weeks**. The hot plate test was performed to assess the thermal nociception levels, and the von Frey test was used to evaluate the limb mechanical pain perception. The methods have been presented (Line 753 to 761, Page 32), and results have also been analyzed as shown below and presented in the “Results” sections in the revised manuscript (Line 305 to 311, Page 15).

Legends (revised Extended Data Fig. 5c-e): Schematic illustration of experimental designs of hot plate test and Von Frey test (left panel). Visualization of the paw withdrawal latency in seconds for the hot plate test for thermal pain sensation (n = 10 at 8 weeks and n = 5 at 12 weeks) (middle panel). Von Frey analysis for nociception shown by stimulus intensity (n = 10 at 8 weeks and n = 5 at 12 weeks) (right panel). Mean values are shown and error bars represent \pm s.d. as analyzed by one-way ANOVA with Tukey’s post hoc tests.

5. A major flaw is the use of a single endpoint in rat and canine models. To keep animal numbers low, this is understandable, but behavioral (functional) evaluation should be assessed over time, not at a single endpoint.

Response: Thanks for this constructive comment. As described above, a few

non-invasive or mini-invasive examinations, such as **sensory function tests (as shown in #4)** and **walking gait analysis (as shown below)**, were performed at postoperative 8 weeks and 12 weeks. In addition, **we also added immunostaining data collected from 4 weeks (observing vascular regrowth and remyelination) and 8 weeks (observing remyelination and axonal outgrowth coupling) to enrich the findings at multiple endpoints.** Therefore, we believe evidence collected at multiple endpoints could better support our conclusion.

Legends (revised Extended Data Fig. 5a, b): Representative images of footprints at postoperative 8 weeks (left), and quantification of sciatic function index (n = 10) (right). Mean values are shown and error bars represent \pm s.d., as analyzed by one-way ANOVA with Sidak's post hoc test.

Legends (revised Extended Data Fig. 5g): Representative immunofluorescence images of

regenerative vessels (upper row) and vessel-myelin coupling (middle row, white arrow indicates representative coupling sites) at postoperative 4 weeks, as well as regenerative axons at 8 weeks (lower row).

The canine nerve defect model was primarily used to evaluate the long-term outcomes and translational significance of SpinMed, and **we supply the examinations including biosafety and effectiveness at postoperative 12 months** in the revised manuscript (detailed data also shown in #2 and #11). The collective results exhibited the superiority of SpinMed in complicated nerve repair.

6. Additionally to point #5 and as stated above in the major summary, a single endpoint well past early regeneration limits the understanding of how these devices affected early regeneration, such as vascular outgrowth and alignment, Schwann cell function and properties, and initial axon outgrowth and coupling.

Response: Thanks for this constructive comment. **We reperformed rat experiments and added new experimental results at postoperative 4 weeks and 8 weeks for illustrating early vascular and neural regeneration** in the revised manuscript. As previous evidence, new vessel outgrowth from nerve stump could be detected within 5 days in case of nerve cut injury (*Cell*, 2015, PMID: 26279190), and vascular extension in an axial alignment manner grows forward at a rate of 200 μm to 250 μm per day as our pilot experiments and previous studies (*J Anat*, 2000, PMID: 11197533; *Bone Res*, 2022, PMID: 35641477). Therefore, we detect new vessel conditions at postoperative 4 weeks and reveal the vascular growth pattern in a rat 15-mm nerve defect model (data shown below).

We also evaluated vessel-myelin coupling at postoperative 4 weeks, and axon outgrowth at postoperative 8 weeks by morphology detection and immunostaining methods (*Cell*, 2015, PMID: 26279190). **The results showed accelerated vascular growth and enhanced vessel-nerve crosstalk at an early stage during neural repair in cases of SpinMed implantation.**

Legends (revised Extended Data Fig. 5f, g): Schematic illustration of cutting planes at postoperative 4 and 8 weeks (left panel), and representative immunofluorescence images of regenerative vessels, vessel-myelin coupling at 4 weeks and extended axons at 8 weeks (right panel).

7. The retrograde labeling results for rodents were confusing. Usually this method allows quantification of neurons (sensory vs motor) regenerating axons, as parent regenerating axons can branch. Therefore, axon counts as an outcome metric alone can lead to results that are misleading, as axons that branch more can lead to an outcome suggesting higher axons counts and thus superior regeneration, when in fact fewer neurons might have regenerated axons. Therefore, the use of this metric by the authors is ideal. However, the methods describe counting neurons, but the results do not report this. Instead, the results describe the use of nanoparticle quantity as a metric, which is misleading and not helpful, as more axon branching could lead to increased accumulation of nanoparticles. It is also unclear what increased nanoparticles means as the methods do not describe how this could be evaluated. The extended data shows neuron counts, but these are much too low of values to reflect neuron quantity regenerating axons in rats (rats have ~20,000 sensory neurons and ~2,000 motor neurons). The methods and results needs attention to understand these data as presented.

Response: Thanks for your kind comments. We are so sorry for the inaccurate descriptions and misleading labels in the initial version, in which some inaccurate

information and confusing expressions relevant to Fluoro-Gold retrograde tracing, especially missed data descriptions in results, appeared in this part.

We have reperformed the axonal retrograde transport experiment according to a standard protocol for completely accurate results (*Nat Neurosci*, 2024, PMID: 38291284; *Biomaterials*, 2013, PMID: 23063298; *Brain*, 2005, PMID: 15872018). In brief, 10 μ l of 4% FluoroGold (FG) solution was injected into the regenerated nerve trunk from the distal terminal. The rats were allowed a survival of 5 days and then anesthetized with an overdose of pentobarbital. The spinal cord segments at L5, L6 and S1, together with the corresponding DRGs, were excised and cut into 20- μ m-thick transverse sections for spinal cords and longitudinal sections for DRGs, followed by observation under a confocal laser scanning microscope with ultraviolet illumination. The number of FG-labeled motor neurons in the anterior horn of spinal cord sections was counted, while the percentage of FG-labeled sensory neurons in DRG sections was calculated by dividing the FG-positive neuron number by the total neuron number. The results even showed more positive neurons were observed in spine cords and DRGs collected from rats in the SpinMed group, and their properties of retrograde axon transport were comparable to those in the autograft group.

These methodology descriptions and main findings have been added to the “Methods” and “Results” sections in the revised manuscript (Line 333 to 341, Page 16; Line 772 to 781, Page 32).

Legends (revised Extended Data Fig. 6h, i): Representative images of FG-labeled neurons observed in DRG sensory neurons or motoneurons within AH (left panel), and quantification for the number of FG-labeled neurons (n = 5) (right panel). Mean values are shown and error bars represent \pm s.d., as analyzed by one-way ANOVA with Tukey's post hoc tests.

8. The experimental design transitioning from the dog vs rat model is confusing and not rationale. Why were different controls used in the rat compared to the dog? The use of branching alone is not a sufficient reason.

Response: Thanks for your kind suggestion. In the present study, we first performed a rat nerve defect model and employed it to evaluate the therapeutic outcomes of SpinMed, as well as underlying vessel-nerve coupling mechanisms. Furthermore, a canine nerve defect model was used to find evidence with higher levels based on satisfactory outcomes observed in rats, owing to the clinically translational significance of SpinMed. In canine models, we aimed to further confirm the biosafety and adaptability of SpinMed with proof of the “precision medicine” concept, and prepare more qualified clues in preclinical trials. **Therefore, the purposes of beagle studies were different from those of rat studies, resulting in distinct experimental designs.**

An individualized hollow bifurcation graft was designed as a SpinMed blank control and implanted into a canine nerve defect model (n = 3), but **it failed to repair critical sciatic nerve defects longer than 4 cm in the case of conduit without multilevel neurium-mimic architectures**, owing to slow neuronal growth. Huge nerve defect and slow restoration, along with serious Wallerian degeneration at the early stage, altogether led to nerve regeneration failure observed from less than 4-month follow-up (*Sci Transl Med*, 2020, PMID: 31969488; *EMBO J*, 2019, PMID: 31268609; *Neural Regen Res*, 2014, PMID: 25206870). During this period, denervation caused right foot ulcers and even incurable tissue defects in beagles in this group, for which the ethics committee recommended we ceased follow-up of beagles in this group and give them euthanasia for relieving pain. Therefore, we did not present related data in the previous version due to inconsistent follow-up periods.

Here we provided some evidence about control individuals as shown by the gross view and TEM images of beagles suffering repair failure in the SpinMed hollow counterpart group at postoperative 16 weeks. These results showed no myelin sheath was detected within regenerative tissue, even though samples were near to proximal stump. It has been also added into the revised figures (revised Extended Data Fig. 7e) to highlight the significance of multilevel neurium-mimic architectures.

Legends (Review Report Fig. 1): Representative images of tissue defect feet (left) of beagles receiving hollow counterpart implantation, and nearly intact feet (right) of beagles in SpinMed group.

Legends (revised Extended Data Fig. 7d, e): Illustration of the local nerve model and

positions of TEM detection (top panel), where transverse ultrastructural views were obtained (bottom panel) (n = 3).

In addition, all of the other 3 groups were direct controls for the SpinMed application. **i)** A single straight counterpart was a geometrical control of SpinMed, the comparison between them revealed **the necessity of geometric guidance and biomimetic reconstruction**. **ii)** The decellularized graft used in this study was a common nerve defect repair strategy by simple suture or cable suture in current clinical practice, at least in China, so it as a commercial product was also a control to help us **evaluate the SpinMed qualities and values**. **iii)** Autograft as the most common positive control in the peripheral nerve studies and gold standard in clinical practice facilitated **determining the level positioning and translational significance of SpinMed**. Collectively, although hollow group was unavailable due to repair failure in beagle experiments that were different from rat study, all of the other groups were designed as various controls for beagle nerve regeneration experiments, globally facilitating the outcome assessment and translational application of SpinMed.

9. The acellular nerve allograft used for the canine is not described and critical to understand. Was it generated from canine or was a human product (AxoGen) used? The methods used to generate this control could yield a product that would not facilitate robust regeneration and could induce an immune response, making it an inaccurate control.

Response: Thanks for your kind suggestion. As we described above, **human-derived decellularized nerves (Nerve Bridge, a commercial product similar to AxoGen) as the state-of-the-art products in our practice and were applied in this study to exhibit repair strategies and outcomes in most clinical scenarios**. Implantation outcome of decellularized grafts (Nerve Bridge) by using cable-suture transplantation into canine nerve defect models was confirmed to be inferior to that by SpinMed implantation. Therefore, **the purpose of this control was to compare SpinMed with products commonly used in clinical practice**.

In the present study, **no evident immune response was found in the decellularized graft group as revealed by a series of biosafety evaluations (revised Supplementary Fig. 11)**, despite a few previous studies reporting that antigenic stimulations induced by decellularized allografts may elicit immune responses. Collectively, the roles of decellularized graft in this study provided a reference to the currently common strategy rather than a blank control, and this comparison highlighted the significance of individualized graft.

10. The dog paradigm of repair is interesting, but not clinically relevant. A sciatic nerve is almost always a cable repair to reflect different branches. The autograft used is not the standard of care-cabled autograft repair is.

Response: Thanks for your kind comment. We are sorry for the unclear descriptions of the surgical procedures of canine models. First, **the defect injury is completely identical in all groups, but repair methods are distinct among experimental groups as shown below.** As illustrated in schemes and surgical images, nerve cable sutures are the primary methods of autograft and decellularized graft implantation. The reasons for this method as follows: **i)** In our clinical practice, sciatic nerve defects are usually critical due to their huge size, for which sural nerves or other nerve donors are acquired and cable-sutured to a robust trunk for autograft. **In this study, the cable method is similar to that in our clinical practice but donor acquired from resected sciatic nerve.** **ii)** If the ipsilateral sural nerve was resected as a donor, we worried about the aberrant sensory function restoration of beagles in the autograft group, which may lead to outcome biases. However, contralateral sural nerve resection may also result in bilateral hind limb disability and abnormal rehabilitation. **Therefore, it was difficult to completely proceed by clinical practice but only simulate it. We determined to reuse the sciatic nerve with branches *in situ*.** **iii)** In the beagle experiments, we also aimed to explore the potential of combined cable suture (proximal) and split suture (distal) *in situ*, despite the similar method has been widely applied to rat autograft models (*Nat Commun*, 2018, PMID: 29358641; *Biomaterials*, 2020, PMID: 32554132).

As confirmed by previous studies, nerve polarity tended nearly not to influence the nerve graft outcomes (*J Reconstr Microsurg*, 2002, PMID: 12177822; *Microsurgery*, 2017, PMID: 27935644), and therefore for autograft in this study, we inverted the resected nerve (180°) to mimic care-cabled autograft for nerve integrity restoration. During this, the epineurium of the distal terminal of resected nerve (branching site, tibial nerve, and common fibular nerve) was partially dissected and side-to-side sutured into a robust nerve trunk, while the proximal sciatic nerve was split along the longitudinal axis to two cables with appropriate ratios for meeting tibial nerve and common fibular nerve stumps respectively (*Neural Regen Res*, 2017, PMID: 29323049; *JAMA Facial Plast Surg*, 2016, PMID: 27197116). After completing cable sutures, the donor cable nerve (preceded by resected nerve) was connected by end-to-end anastomosis with stumps. We believe this autograft model is appropriate for the present beagle experiment.

Legends (revised Fig. 6a, b): Illustration of surgical procedures performed in complicated sciatic nerve defects in the “Single” “SpinMed” “Decellularized graft” and “Autograft” groups (a), and surgical images of application methods in various groups, with labels for sciatic nerve (1), tibial nerve (2) and common peroneal nerve (3).

11. The dog studies do not provide sufficient methodological detail. The experiments are not the same as rodent, where the functional (behavioral) assay is completely different. Without more detail, this cannot be assessed. As well, it is unclear why the histology in this model was only partially quantified. Axon quantity should be

reported.

Response: Thanks for your kind suggestion. We are sorry for the insufficient methodological details about canine studies in the initial version. In the previous manuscript, distinct objectives and highlights led to differential data presentations in canine and rodent studies, as well as separate figures displayed in the main text and Supplementary Materials. To address these concerns, we have revised this part as follows.

First, we have provided the modeling scheme and surgical images for each group as shown above (#10) and the revised manuscript. The illustration and added methods (Line 732 to 739, Page 31) could contribute to a better understanding of canine surgical procedures.

Furthermore, we added electrophysiology assessment results at postoperative 16 weeks, 8 months, and 12 months, for which the relevant methods and findings were shown below and in the “Results” section (Line 376 to 379, Page 18).

Legends (revised Extended Data Fig. 7c): Quantitative analysis of nerve conduction velocity and CMAP amplitude recovery index at 16 weeks (n = 3). Mean values are shown and error bars represent \pm s.d., as analyzed by one-way ANOVA with Tukey’s post hoc tests.

Legends (revised Fig. 6c): Analysis of nerve conduction velocity and CMAP amplitude recovery index by electrophysiology assessment at postoperative 8 and 12 months (n = 3).

Mean values are shown and error bars represent \pm s.d., as analyzed by one-way ANOVA with Tukey's post hoc tests.

We have reorganized the TEM images of regenerative nerve in several groups at postoperative 16 weeks followed by more parameter quantification, such as sheath thickness and g-ratio. In addition, we added TEM data of regenerative nerves acquired from samples of postoperative 12 months into the revised manuscript, also shown below.

Legends (revised Extended Data Fig. 7e, f): Representative TEM images of regenerative nerves collected from P1 to P4 as described above (top panel). Quantitative analysis for myelin sheath thickness and g-ratio within P2 to P4 at postoperative 16 weeks ($n = 3$, Holl. group not in quantification). Mean values are shown and error bars represent \pm s.d., as analyzed by one-way ANOVA with Tukey's post hoc tests.

Legends (revised Fig. 6d-g): Illustration of the sample collection positions at postoperative 12 months for TEM detection (left panel), where transverse ultrastructural views were obtained (right panel) (n = 3). Quantification of parameters for the nerve ultrastructural analysis at the P3 and P4, including the myelin sheath thickness and the g-ratio (lower panel) (n = 3). Mean values are shown and error bars represent \pm s.d., as analyzed by one-way ANOVA with Tukey's post hoc tests.

Finally, the canine nerve and muscle histology followed by parameter quantification at postoperative 12 months were also provided in the revised manuscript (as shown below), which undoubtedly made the long-term results solid (Line 392 to 397, Page 18).

Legends (revised Fig. 6h, i): Representative images of H&E (left panel, upper row) and toluidine blue (left panel, lower row) staining for regenerative nerves collected from the P2 at postoperative 12 months (n = 3), and quantification of the number of myelinated nerve fibers (right panel). Mean values are shown and error bars represent \pm s.d., as analyzed by one-way ANOVA with Tukey's post hoc tests.

Legends (revised Supplementary Fig. S15): Representative images of gastrocnemius Masson staining at postoperative 12 months (upper panel) followed by quantification for fiber area (low panel, left) and collagen volume (low panel, right) (n = 3). Mean values are shown and error bars represent \pm s.d., as analyzed by one-way ANOVA with Tukey's post hoc tests.

12. The statistics is unclear. ANOVA is used for multiple groups, but no post-hoc methods are provided.

Response: Thanks for your kind suggestion. We are sorry for missing the descriptions of post-hoc tests. **Comparations among multiple groups (three groups or more) in this study were performed by using one-way ANOVA followed by Tukey's post-hoc test unless otherwise stated.** According to it, we also re-checked all statistical outcomes and *p* values in revised figures and manuscripts. The modified descriptions have been added to the revised manuscript (Line 804 to 805, Page 33).

Format revisions:

We have double-checked all texts and figure formats according to the requirements of “Formatting Instructions” at the website of “Nature Communications.”

We hope this version can meet the acceptance standards. Thank you!

REVIEWERS' COMMENTS

Reviewer #1 (Remarks to the Author):

The authors did a great job in revision, and have addressed all my previous comments. The manuscript is recommended for publication.

Reviewer #2 (Remarks to the Author):

The authors have done an excellent job to resolve comments and provide additional supporting detail and data. The manuscript is now acceptable.

Reviewer #1 (Remarks to the Author):

The authors did a great job in revision, and have addressed all my previous comments.

The manuscript is recommended for publication.

Response: Thank you very much for the positive comment and evaluation for our revision work.

Reviewer #2 (Remarks to the Author):

The authors have done an excellent job to resolve comments and provide additional supporting detail and data. The manuscript is now acceptable.

Response: Thank you very much for the previously constructive comments and kind recommendations.